# Reinforced Cross-Domain Knowledge Distillation on Time Series Data

**Qing Xu**
Institute for Infocomm Research
A*STAR, Singapore
Nanyang Technological University
Xu_Qing@i2r.a-star.edu.sg

**Min Wu**
Institute for Infocomm Research
A*STAR, Singapore
wumin@i2r.a-star.edu.sg

**Xiaoli Li**
Institute for Infocomm Research, A*STAR, Singapore
A*STAR Centre for Frontier AI Research, Singapore
xlli@i2r.a-star.edu.sg

**Kezhi Mao**
Nanyang Technological University
EKZMao@ntu.edu.sg

**Zhenghua Chen**[*]
Institute for Infocomm Research, A*STAR, Singapore
A*STAR Centre for Frontier AI Research, Singapore
chen0832@e.ntu.edu.sg

## Abstract

Unsupervised domain adaptation methods have demonstrated superior capabilities in handling the domain shift issue which widely exists in various time series tasks. However, their prominent adaptation performances heavily rely on complex model architectures, posing an unprecedented challenge in deploying them on resource-limited devices for real-time monitoring. Existing approaches, which integrates knowledge distillation into domain adaptation frameworks to simultaneously address domain shift and model complexity, often neglect network capacity gap between teacher and student and just coarsely align their outputs over all source and target samples, resulting in poor distillation efficiency. Thus, in this paper, we propose an innovative framework named **R**einforced **C**ross-**D**omain **K**nowledge **D**istillation (**RCD-KD**) which can effectively adapt to student's network capability via dynamically selecting suitable target domain samples for knowledge transferring. Particularly, a reinforcement learning-based module with a novel reward function is proposed to learn optimal target sample selection policy based on student's capacity. Meanwhile, a domain discriminator is designed to transfer the domain invariant knowledge. Empirical experimental results and analyses on four public time series datasets demonstrate the effectiveness of our proposed method over other state-of-the-art benchmarks. Our source code is available at https://github.com/xuqing88/Reinforced-Cross-Domain-Knowledge-Distillation-on-Time-Series-Data.

## 1 Introduction

Recent years have witnessed great successes of deep neural networks (DNNs) in various time series applications [1; 2; 3; 4]. Nevertheless, a significant drawback impeding their scalability is the limited

---

[*]Corresponding author

38th Conference on Neural Information Processing Systems (NeurIPS 2024).

generalization capability on unseen data. This challenge arises when there is a distribution disparity between the data used for training and deployment. For instance, a fault diagnosis model trained on certain machines may perform poorly on the data collected from other machines which have different working conditions and configurations. Collecting and annotating data for each machine would be very laborious and costly. To handle this, various unsupervised domain adaptation (UDA) methods have been extensively explored. These methods aim to transfer the domain invariant knowledge from an existing labeled data domain (*i.e.*, *source domain*) to an unlabeled domain (*i.e.*, *target domain*) either by explicitly minimizing certain pre-defined discrepancy metrics [5; 6] or implicitly learning domain-invariant representations with adversarial manners [7; 8]. However, these UDA methods heavily rely on the complex network architectures and their adaptation performance will significantly degrade with shallower networks [9; 10]. The over-parameterized DNNs will inevitably lead to another practical issue in industries. For many real-world time series tasks, the developed models are often required to be deployed on edge devices with very limited computational resources, such as smartphones and robots, for real-time and long-term monitoring. The intolerable computational and storage burdens make the deployment of those complex DNNs on edge devices become an unprecedented challenge.

Some pioneering efforts have been made to integrate knowledge distillation (KD) techniques into UDA frameworks to transfer the cross-domain knowledge from a cumbersome teacher to a compact student for the reduction of model complexity. However, we empirically find that simply integrating KD with UDA frameworks like existing works will make the compact student suffer from unsatisfying adaptation performance. The rationale behind this lies in the facts that: on the one hand, due to its limited network capacity, the compact student may fail to capture the same fine-grained patterns in data as the cumbersome teacher. Coarsely aligning its feature representations or outputs with the teacher like [11; 12] will impede its learning process and result in sub-optimal performance on the target domain. On the other hand, in the cross-domain scenario, teacher's knowledge on each individual target sample may not be always reliable and instructive due to the lack of label supervision in target domain. Blindly trusting teacher's knowledge for all samples, especially on target domain, will result in negative transfer. Therefore, to achieve good adaptation performance on the target domain, we have to adaptively transfer teacher's knowledge based on student's network capability.

Motivated by above insights, we propose a novel end-to-end framework for cross-domain knowledge distillation to simultaneously address domain shift and model complexity. To be specific, an adversarial discriminator module is designed to align teacher's and student's representations between source and target domains on latent feature space for domain-invariant knowledge transfer. Meanwhile, to adaptively transfer teacher's knowledge on the unlabeled target domain, we formulate the target sample selection problem under a reinforcement learning framework. For a specific target sample, if the student demonstrates the ability to attain the same uncertainty level as the teacher (*i.e.*, uncertainty consistency), or can largely mimic teacher's outputs (*i.e.*, sample transferability), we deem such a sample suitable for knowledge distillation. Based on that, we design a novel reward function according to student's learning capability. A dueling Double Deep Q-Network (DDQN) is then utilized to learn the optimal target sample selection policy for mitigating the negative effects of unsuitable knowledge from teacher. Our contributions are summarized as follows:

- An end-to-end framework named **R**einforced **C**ross-**D**omain **K**nowledge **D**istillation (**RCD-KD**) is proposed to not only effectively transfer the domain-invariant knowledge but also dynamically distill the adaptive target knowledge based on student's learning capability.

- We develop an innovative reinforcement learning-based module to learn the optimal target sample selection policy for robust knowledge distillation. A novel reward function is designed for assessing student's learning capability in terms of uncertainty consistency and sample transferability to dynamically transfer teacher's target knowledge.

- The extensive experimental results on four real-world time series tasks demonstrate the superior effectiveness of our approach compared to other SOTA methods.

## 2  Related Work

In recent years, there are some pioneering works to tackle both domain shift and model complexity simultaneously. A resource efficient domain adaptation (REDA) framework with multi-exit architectures is proposed in [9], where the 'easier' samples are inferred via early exits and 'harder' ones are

inferred via top exit. Meanwhile, some other researchers leverage the knowledge distillation [13] to enhance the adaptation performance of the compact student. For instance, a framework named knowledge distillation for unsupervised single target domain adaptation (KD-STDA) is proposed in [11]. Teacher's knowledge is gradually transferred via dynamically adjusting the contributions of UDA and KD loss. Similarly, a multi-level distillation for Domain Adaptation (MLD-DA) strategy is proposed in [12] to improve the distillation efficiency via a novel cross entropy loss. However, the above two methods transfer the knowledge from both source and target domains. We empirically show that the source domain-specific knowledge might have negative contribution to student's generalization. Besides, MobileDA [14] and adversarial adaptation with distillation (AAD) [15] employ the teacher trained on source-only domain to guide student's training, which have already been proved inefficient due to the limited and biased knowledge from teacher model by [4]. Moreover, to achieve more reliable knowledge from teacher, in [16] a maximum cluster difference metric is proposed to estimate teacher's confidence on certain sample. In [4], a framework named universal and joint knowledge distillation (UNI-KD) is proposed to measure teacher's confidence on individual sample via the output of a data-domain discriminator. However, due to the compact network architecture of the domain-shared feature extractor from student, the estimated uncertainty is not reliable. In our work, we estimate teacher's knowledge with student's capacity and then utilize it as the reward for the learning process of RL-based target sample selection module. The experimental results demonstrate that our proposed method can better enhance student's performance on target domain. Meanwhile, our work also relates to active learning (AL) field specifically in terms of selecting the most critical instances from unlabeled data. Note that here we only discuss the uncertainty-based sampling strategies in active learning as other query strategies (e.g., instance correlation) are beyond the scope of our paper. In AL, the uncertainty can be measured by three metrics: least confidence [17; 18], sample margin [19], and sample entropy [20]. Particularly, the entropy metric measures the uncertainty over the whole output prediction distribution [21; 22]. In our method, instead of explicitly utilizing entropy-based uncertainty as AL methods, we leverage the consistency between teacher' and student's entropy-based uncertainty to learn the optimal sample selection policy with dueling DDQN. See **Supplementary** for more comparison results.

## 3    Methodology

### 3.1    Preliminaries

Following standard UDA setup, we consider data from two domains: a labeled *source* domain $\mathcal{D}_{src}^L = \{(x_s^i, y_s^i)\}_{i=1}^{n_s}$ and an unlabeled *target* domain $\mathcal{D}_{tgt}^U = \{x_t^i\}_{i=1}^{n_t}$ which shares the same label space as source domain but has different data distributions. Here, $n_s$ and $n_t$ are the number of training samples in source and target domains, respectively. A powerful teacher model $T$ with superior adaptation performance is first pre-trained on $\mathcal{D}_{src}^L \bigcup \mathcal{D}_{tgt}^U$ with SOTA UDA methods. Our objective is to train a compact student model $S$ which is not only shallower than the teacher model but also can achieve competitive performance on unlabeled target domain. To transfer the learned knowledge from teacher to student, one can just follow standard KD [13] and force the student to mimic teacher's soften logits via Eq. (1). Here, $\mathcal{X}_b$ represents a batch of training samples and KL refers to the Kullback–Leibler divergence. $\boldsymbol{q}^S$ and $\boldsymbol{p}^T$ are the softmax outputs soften by a temperature factor $\tau$ from student $S$ and teacher $T$, respectively. They are calculated by $q_c^S = exp(z_c/\tau)/\sum_{j=1}^C exp(z_j/\tau)$, where $C$ is the number of classes and $q_c^S$ represents the student's prediction probability of a certain sample belonging to the $c$-th class.

$$\mathcal{L}_{KD} = \sum_{x \in \mathcal{X}_b} KL(\boldsymbol{p}^T || \boldsymbol{q}^S) = \sum_{x \in \mathcal{X}_b} \sum_c p_c^T log(p_c^T / q_c^S). \tag{1}$$

However, since the teacher is trained on unlabeled target data, its prediction performance on specific target sample cannot be guaranteed. The compact architecture of student also limits its ability to fully accept teacher's knowledge. In other words, directly minimizing the distribution discrepancy between teacher's and student's predictions over all target samples might introduce inappropriate knowledge which will mislead the student's learning process. Thus, we propose to alleviate the above issue with a novel RL-based target sample selection module which can dynamically select suitable samples to assist the knowledge transferring. Fig. 1 illustrates the details of our proposed method.

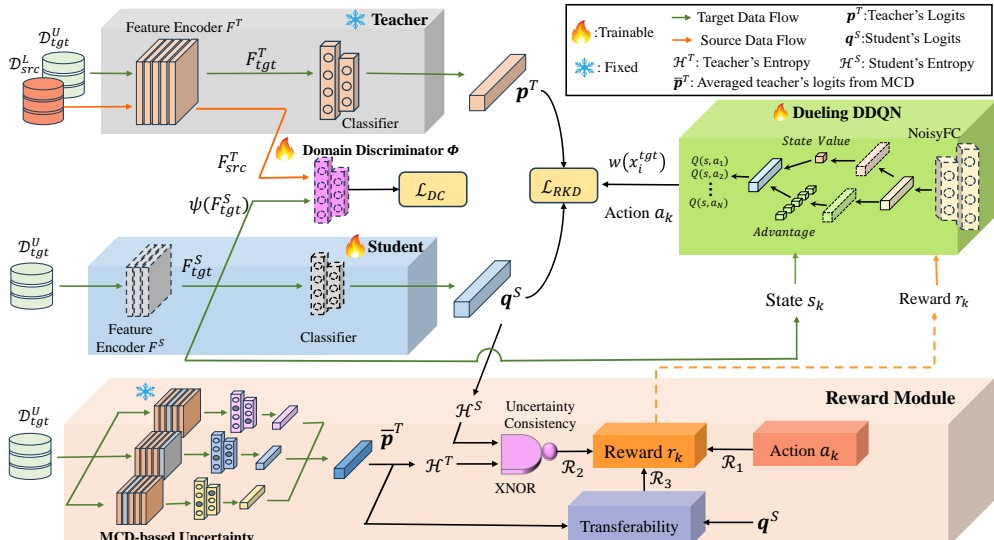

Figure 1: Illustration of proposed RCD-KD. A Monte Carlo Dropout (MCD) based reward module is utilized to generate the reward for learning the optimal target sample selection policy. Specifically, the reward function consists of three parts. The first one is the action $a_k$ which is the output of dueling DDQN. The second part is the uncertainty consistency, estimated by entropy from student's logits $q^S$ and the averaged logits $\overline{p}^T$ of $N$ teachers generated from MCD module. The third part is the sample transferability based on the KL divergence between $q^S$ and $\overline{p}^T$. The output of reward module $r_k$ then will be utilized for the optimization of dueling DDQN for learning optimal sample selection policy. Meanwhile, a domain discriminator $\Phi$ is employed to transfer the domain-invariant knowledge.

## 3.2 RL-based Target Sample Selection

Following [23; 24], we consider target sample selection task as a Markov Decision Process which can addressed by reinforcement learning. A RL-based target sample selection module is first designed to enhance the distilling efficiency of teacher's knowledge on target domain. Particularly, a dueling DDQN [25] is employed to learn the optimal target sample selection policy. The dueling architecture can effectively mitigate the risk of overestimation by separately estimating the state value and advantage function, which improves the accuracy of action-value predictions. Meanwhile, to tackle the instability issue often encountered in training deep reinforcement learning models, we leverage strategies such as target network and experience replay. Specifically, the target network provides more stable targets for updating the Q-values by maintaining a separate, slowly updated network for generating target values, while experience replay enables the model to learn from a diverse set of past experiences, further enhancing stability and convergence during training. In each training batch, we utilize the learned sample selection policy to adaptively transfer teacher's target knowledge according to student's learning capability. In the following, the detailed definition of state, action, reward and the optimization of dueling DDQN will be introduced.

**State.** Given a batch of target domain samples $\{x_i^{tgt}\}_{i=1}^{n_b}$ and student's feature extractor $F^S$, the state $s_k$ at episode $k \in [1, K]$ is defined as the feature representations from student's feature extractor $s_k = [F_k^S(x_1^{tgt}), ..., F_k^S(x_{n_b}^{tgt})]$. Here, $n_b$ is the batch size and $K$ represents the maximum episode length. The state $s_k$ is forwarded to the dueling DDQN, generating a set of actions that decide whether to retain or discard the corresponding target samples. The student is subsequently optimized with the selected samples, and the next state $s_{k+1}$ can be obtained with the updated student. A terminate state will be triggered if the target sample is not selected at time step $k$ or the episode reaches the maximum episode length $K$.

**Action.** For a specific target sample $x_i^{tgt}$ in a batch, it only has two actions $a_i \in \{0, 1\}$ which is binary. Specifically, $a_i = 1$ means to retain the sample and $a_i = 0$ means to discard it. Since the output of dueling DDQN parameterized with $\Theta_q$ is a two-dimensional $Q$-value vector, the optimal

action $a_i^*$ for sample $x_i^{tgt}$ at current episode $k$ thus can be calculated by Eq. (2). Subsequently, the binary weights for all target samples are formulated as $w = [a_1^*, ..., a_{n_b}^*]$, which then are utilized to calculate the distillation loss $\mathcal{L}_{RKD}$.

$$a_i^* = \underset{a}{\operatorname{argmax}} Q(F_k^S(x_i^{tgt}), a; \Theta_q). \tag{2}$$

**Reward.** The reward function is pivotal in shaping the learning process for target sample selection policy, as it offers essential feedback to DDQN regarding the value associated with selecting a specific action in the current state. To achieve reliable knowledge transferred from selected target samples, we propose to utilize model uncertainty and sample transferability to design the reward function.

The first component constructing our reward function is a Boolean function $\mathcal{R}_1 = (a_i == 1)$, which indicates whether the sample $x_i$ is retained or not. The second component of our reward function is called uncertainty consistency reward. The motivate is straightforward: due to the lacks of label in target domain, we expect that the student should have the same uncertainty level as the teacher for a specific sample $x_i$. For the teacher model, we employ the Monte Carlo Dropout (MCD) [26] to estimate its uncertainty, which utilizes a dropout distribution to approximate the posterior distribution (See **Supplementary** for more details). Practically, it means to enable the dropout of teacher model and forward $N$ times for each sample $x_i$ and the averaged prediction $\overline{\boldsymbol{p}}_i^T = \frac{1}{N} \sum_n p(y = c|x_i, \boldsymbol{\theta}^n)$ can be utilized to calculate the entropy $\mathcal{H}_i^T = -\sum_c \overline{p}_{i,c}^T log(\overline{p}_{i,c}^T)$ for measuring its uncertainty. Here, $p(y = c|x_i, \boldsymbol{\theta}^n)$ represents the probability of sample belonging to class $c$ and it is the softmax outputs of teacher model on the $n$-th forward pass. For the student, the uncertainty is calculated with $\mathcal{H}_i^S = -\sum_c q_{i,c}^S log(q_{i,c}^S)$. Intuitively, a higher value of the predictive entropy $\mathcal{H}$ will be obtained when all classes are predicted to have equal probabilities, which means the model is less confident about the specific data. To ensure the student has consistent uncertainty level as the teacher, we formulate the uncertainty consistency reward $\mathcal{R}_2 = (\mathcal{H}_i^S > \frac{1}{n_b} \sum_{j=1}^{n_b} \mathcal{H}_j^S) \odot (\mathcal{H}_i^T > \frac{1}{n_b} \sum_{j=1}^{n_b} \mathcal{H}_j^T)$, where $\odot$ is the exclusive-nor operation. The third component of our reward function is the transferability reward formulated as $\mathcal{R}_3 = (\mathcal{D}_i < \frac{1}{n_b} \sum_{j=1}^{n_b} \mathcal{D}_j)$. Here, $\mathcal{D}_i = KL(\overline{\boldsymbol{p}}_i^T || \boldsymbol{q}_i^S)$ is the KL divergence between student's prediction and the averaged MCD teacher prediction. Apparently, the samples whose KL divergence are lower than the averaged divergence are easier ones for the compact student to learn. With the above three Boolean functions, our reward function is defined as Eq. (3) shows:

$$r_k = \alpha_1 * (\mathcal{R}_1 \oplus \mathcal{R}_2 - 0.5) + \alpha_2 * (\mathcal{R}_1 \oplus \mathcal{R}_3 - 0.5), \tag{3}$$

where $\oplus$ is the exclusive-or operation. For the first part of Eq. (3), a positive reward value will be assigned if a sample is retained and student shows consistent uncertainty as the teacher, or it is discarded and student and teacher show inconsistent uncertainty about it. Otherwise, a negative reward will be assigned. Similarly, for the second part of Eq. (3), a positive reward will be assigned if its transferability is higher than others and being selected, or its transferability is lower than others and not selected. We utilize $\alpha_1$ and $\alpha_2$ to adjust the contribution of each part. We constrain the reward within the range of -1 to 1 to offer explicit guidance to the DDQN so that it can efficiently learn to distinguish between good and bad actions.

**Dueling DDQN Optimization.** The dueling deep Q-network consists of two streams as shown in Fig. 1: state-value estimation stream $V(s; \Theta_E, \Theta_V)$ parameterized with $\Theta_E$ and $\Theta_V$ and advantages estimation stream $A(s, a; \Theta_E, \Theta_A)$ for each action which is parameterized with $\Theta_E$ and $\Theta_A$. Here, $\Theta_E$ is a shared encoder. Furthermore, to balance the exploitation and exploration, we adopt the NoisyNet [27] for the fully-connected layers in $\Theta_E$, $\Theta_V$ and $\Theta_A$. Besides, a replay buffer $\mathcal{M}$ is designed to store the historical experience $(s_k, a_k, r_k, s_{k+1}, d)$, where $d \in \{0, 1\}$ indicates whether the next step $k + 1$ is the terminal step ($d = 0$) or not ($d = 1$). A batch of entries in $\mathcal{M}$ will be randomly sampled out for DDQN optimization.

To train the dueling DDQN (*i.e.*, the online network $\mathcal{Q}$), another target Q-network $\mathcal{Q}'$ is desired, which has identical network architecture as $\mathcal{Q}$ but is optimized in a different way. Specifically, the online network $\mathcal{Q}$ is to estimate the Q-values $Q_{est}$ by aggregating two steams via Eq. (4):

$$Q_{est} = V(s; \Theta_E, \Theta_V) + A(s, a; \Theta_E, \Theta_A) - \frac{1}{2} \sum_{a_i} A(s, a_i; \Theta_E, \Theta_A). \tag{4}$$

The target Q-network $\mathcal{Q}'$ is to generate the target Q-values as Eq. (5) shows. Here, $\Theta = \{\Theta_E, \Theta_V, \Theta_A\}$, $\Theta' = \{\Theta'_E, \Theta'_V, \Theta'_A\}$ are the parameters of $\mathcal{Q}$ and $\mathcal{Q}'$, respectively. $\gamma \in [0, 1]$ is the discount factor to balance the immediate and future rewards.

$$Q_{tar} = r_k + d * \gamma * Q(s_{k+1}, \underset{a_{k+1}}{\operatorname{argmax}} Q(s_{k+1}, a_{k+1}; \Theta); \Theta'). \tag{5}$$

The online network $\mathcal{Q}$ is optimized by minimizing the Huber loss between $Q_{est}$ and $Q_{tar}$. The target network $\mathcal{Q}'$ is updated with a moving average method as shown in Eq. (6), where $\delta$ is a smoothing parameter determining how much historical information of the online network to be transferred to the target network.

$$\Theta' \leftarrow \delta * \Theta' + (1 - \delta) * \Theta. \tag{6}$$

### 3.3 Student Optimization

With the proposed RL module, we can efficiently transfer adaptive knowledge from the teacher model to the student model by dynamically eliminating target samples which are unsuitable for student learning. Particularly, we reformulated Eq. (1) to Eq. (7), where $w = [a_1^*, ..., a_{n_b}^*]$ is the output of online Q-network $\mathcal{Q}$. By minimizing $\mathcal{L}_{RKD}$, student's generalization capability on target domain can be effectively enhanced.

$$\mathcal{L}_{RKD} = \sum_{x \in \mathcal{X}_b} w_j * \sum_i p_i^T log(p_i^T / q_i^S). \tag{7} \qquad \mathcal{L}_{DC} = -\mathbb{E}[log(\Phi(\psi(F^S(x_{tgt}))))]. \tag{8}$$

Meanwhile, to transfer the domain-invariant knowledge between two domains from teacher to student, we design an adversarial leaning-based module as depicted in Fig. 1, followed [28]. Particularly, a domain discriminator $\Phi$ is employed to distinguish the source of input feature maps (*i.e.*, whether the feature maps are generated from the teacher with source data as inputs or the student with target data as inputs). Since the dimensions of student's and teacher's feature maps are different, an adaptor layer $\psi$ is employed to match their dimensions. The domain confusion loss is then formulated as Eq. (8). It is worth noting that in our experiments, we utilize the DANN [7] to pre-train the teacher. Although other DA methods can also be adopted in our framework, the DANN can essentially provide a pre-trained accurate domain discriminator after teacher's training. During transferring domain-invariant knowledge, we can re-utilize it and only optimize the student and the adaptor layer $\psi$, which will significantly improve the training efficiency. Meanwhile, it is also possible that one may utilize some other UDA methods to pre-train the teacher. In this case, the domain discriminator $\Phi$ has to be adversarially trained against the student by minimizing loss $\mathcal{L}_{adv}$ as Eq. (9). More experimental results in terms of utilizing other UDA methods to train the teacher can be found in Experiments section.

$$\mathcal{L}_{adv} = -\mathbb{E}[log\Phi(F^T(x_{src}))] - \mathbb{E}[log(1 - \Phi(\psi(F^S(x_{tgt}))))]. \tag{9}$$

The overall loss for student optimization is calculated via Eq. (10). $\lambda$ is a hyperparameter to balance the contribution of each part. Algorithm 1 shows details of proposed **RCD-KD**.

$$\mathcal{L} = \mathcal{L}_{DC} + \lambda * \tau^2 * \mathcal{L}_{RKD}. \tag{10}$$

## 4   Experiments

### 4.1   Experimental Setup

**Datasets.**   To evaluate our method, extensive experiments are conducted on four public datasets across three different tasks, namely human activity recognition, rolling bearing fault diagnosis and sleep stage classification. To be specific, the first dataset is called human activity recognition (**HAR**) [29] for identifying subject's activities (*i.e,*, *walk*, *walk upstairs*, *walk downstairs*, *stand* and *sit*). Sensory measurements from the accelerometer and gyroscope embedded in a smartphone were

**Algorithm 1** Proposed **RCD-KD**

---

**Input**: Teacher $T$, Student $S$, adaptation model $\psi$, domain discriminator $\Phi$, online and target Q-network $\mathcal{Q}$ and $\mathcal{Q}'$, source data $\mathcal{D}_{src}^L$, target data $\mathcal{D}_{tgt}^U$ and Replay buffer $\mathcal{M}$

1: **for** every epoch **do**
2:    **for** every batch $X_{tgt} \in \mathcal{D}_{tgt}^U$ and $X_{src} \in \mathcal{D}_{src}^L$ **do**
3:       **for** episode $k \in [1, K]$ **do**
4:          Get state $s_k$, sample action $a_k \sim \mathcal{Q}(s_k)$ and update next state $s_{k+1}$
5:          Update $S$ and $\psi$ by minimizing $\mathcal{L}$ as Eq. (10)
6:          Calculate $r_k$ via Eq. (3); Set $d = 0$ if episode end, otherwise $d = 1$
7:          Store $(s_k, a_k, r_k, s_{k+1}, d)$ to $\mathcal{M}$
8:          **if** $\Phi$ is not pre-trained **then**
9:             Fix the parameters in $S$ and $\psi$ and update $\Phi$ via minimizing $\mathcal{L}_{adv}$ in Eq. (9)
10:     Sample a random batch $(s_k, a_k, r_k, s_{k+1}, d)$ from $\mathcal{M}$
11:     Calculate $Q_{est}$ via Eq. (4) and $Q_{tar}$ via Eq. (5), update $\mathcal{Q}$ via Huber loss and $\mathcal{Q}'$ via Eq. (6)

---

Table 1: Performance comparison with other UDA methods.

| Datasets | Student-Only | Metric-based | | | | Adversarial-based | | Ours |
|---|---|---|---|---|---|---|---|---|
| | | HoMM [6] | MDDA [5] | SASA [36] | DANN [7] | CoDATS [30] | AdvSKM [32] | |
| HAR | 55.94±8.99 | 83.62±1.82 | 84.89±6.29 | 83.37±3.23 | 82.42±3.82 | 75.72±8.62 | 70.72±4.06 | **94.68±1.62** |
| HHAR | 58.74±10.79 | 68.02±6.59 | 73.26±8.35 | 77.13±4.09 | 76.03±1.97 | 74.64±4.18 | 63.24±5.99 | **82.37±1.84** |
| FD | 66.78±4.38 | 74.52±6.00 | 81.80±5.43 | 86.75±2.39 | 77.95±8.52 | 77.54±9.45 | 77.83±5.71 | **92.63±0.62** |
| SSC | 50.39±7.67 | 59.79±5.51 | 57.45±3.68 | 59.36±3.69 | 57.39±5.51 | 57.21±5.61 | 57.28±4.77 | **67.49±1.83** |

collected from 30 subjects. Considering the variability among subjects, each subject is considered as a single domain and the adaptation is performed between two subjects. Here, we follow existing works [30; 4] and select five transfer scenarios. The second evaluation dataset is Heterogeneity human activity recognition (**HHAR**) [31]. Compared to **HAR**, the sensory measurements are collected with various models of smartphones from different manufacturers which are positioned with various orientations on subjects. Thus, the domain gaps between different subjects are generally considered to be larger than **HAR**. Five transfer scenarios are selected for evaluation same as previous work [32]. The third dataset is rolling bearing fault diagnosis (**FD**) [33] which aims to classify the health status of rolling bearing from *healthy*, *artificial damages*, *damages from accelerated lifetime tests*. The rolling bearing are tested under various operation conditions. Same as [4; 34], five transfer scenarios between different operational configurations are selected for fair comparison. The last evaluation dataset is sleep stage classification (**SSC**) dataset [35], which intends to recognize subject's sleep stages (*i.e.*, *wake, non-rapid eye movement N1, N2, N3 and rapid eye movement stage*) with electroencephalography waveform. Five scenarios are evaluated following previous study [34].

**Implementations.** For the proposed RL-based sample selection module, we set $\gamma = 0.9$ and $\delta = 0.999$ following [25] in Eq. (5) and (6), respectively. We set $N = 10$ to calculate teacher's entropy for the reward function and $K = 5$ for the episodes to generate historical experience. Note that to guarantee fair comparison, we ensure the total training steps of ours and benchmark methods are same. Furthermore, we adopted the **1D-CNN** as the backbone of the teacher and student models following [4; 34], where student is a shallow version of teacher with less filters (See **Supplementary** for detailed network architectures of $T$, $S$, $\Phi$ and dueling DDQN). For $\alpha_1$, $\alpha_2$ in Eq. (3) and $\lambda$, $\tau$ in Eq. (10), we use the grid search and set $\alpha_1 = 0.2$, $\alpha_2 = 1.8$, $\lambda = 1.0$, $\tau = 2$ for all experiments. More sensitivity analysis regarding $N$, $K$, $\lambda$, $\alpha_1$, $\alpha_2$ and $\tau$ can be found in **Supplementary**. The averaged macro F1-score with three independent running is reported.

## 4.2 Benchmark with UDA methods

To demonstrate the effectiveness of our proposed RCD-KD, we first compare it with other advanced UDA methods as shown in Table 1. Note that all of the benchmark UDA methods are directly applied to the compact student. From Table 1, some observations can be found. In most transfer scenarios, directly applying UDA methods (either the metric-based or adversarial-based) can improve the performance of compact student model on target domain. However, these methods perform inconsistently across different tasks. For instance, HoMM performs best on **SSC**, but worst on **FD** compared to other UDA methods. Meanwhile, the improvement of these UDA methods is

Table 2: Marco F1-scores on HAR and HHAR across three independent runs.

| Methods | HAR Transfer Scenarios | | | | | | HHAR Transfer Scenarios | | | | | |
|---|---|---|---|---|---|---|---|---|---|---|---|---|
| | 2→11 | 6→23 | 7→13 | 9→18 | 12→16 | Avg | 0→6 | 1→6 | 2→7 | 3→8 | 4→5 | Avg |
| Teacher | 100.0 | 100.0 | 99.92 | 93.69 | 81.65 | 95.05 | 64.47 | 94.23 | 57.22 | 98.88 | 97.69 | 82.50 |
| Student-Only | 68.51 | 59.57 | 78.88 | 21.02 | 51.71 | 55.94 | 50.46 | 65.95 | 43.22 | 58.84 | 75.22 | 58.74 |
| KD-STDA [11] | 98.31 | 89.55 | 89.28 | 67.41 | 63.13 | 81.54 | 46.15 | 92.19 | 41.69 | 96.51 | 89.79 | 73.27 |
| KA-MCD [16] | 89.46 | 59.26 | 63.62 | 58.93 | 45.67 | 63.39 | **65.25** | 90.59 | 42.57 | 85.71 | 85.48 | 73.92 |
| MLD-DA [12] | **100.0** | 99.11 | 92.96 | 82.78 | 64.08 | 87.79 | 61.53 | 94.32 | 47.91 | 91.07 | 92.74 | 77.51 |
| REDA [9] | 99.44 | 93.81 | 92.43 | 74.55 | 55.77 | 83.20 | 32.05 | 93.85 | 36.10 | 90.24 | 95.41 | 69.53 |
| AAD [15] | 83.74 | 90.89 | 83.05 | 75.96 | 61.67 | 79.06 | 53.25 | 81.22 | 48.35 | 87.00 | 86.36 | 71.24 |
| MobileDA [14] | 92.71 | 90.19 | 91.39 | 77.95 | 64.34 | 83.32 | 46.60 | 93.31 | 49.13 | 98.30 | 96.84 | 76.84 |
| UNI-KD [4] | **100.0** | 96.33 | 93.20 | 79.77 | 64.91 | 86.84 | 46.66 | **94.89** | **59.20** | 98.45 | **97.42** | 79.32 |
| **Ours** | **100.0** | **100.0** | **99.64** | **92.87** | **80.87** | **94.68** | 64.47 | 94.24 | 57.59 | **98.45** | 97.11 | **82.37** |

Table 3: Marco F1-scores on FD and SSC across three independent runs.

| Methods | FD Transfer Scenarios | | | | | | SSC Transfer Scenarios | | | | | |
|---|---|---|---|---|---|---|---|---|---|---|---|---|
| | 0→1 | 0→3 | 2→1 | 1→2 | 2→3 | Avg | 0→11 | 12→5 | 16→1 | 7→18 | 9→14 | Avg |
| Teacher | 88.36 | 86.46 | 88.82 | 99.84 | 99.92 | 92.68 | 51.43 | 68.71 | 73.48 | 72.48 | 76.59 | 68.54 |
| Student-Only | 34.94 | 42.14 | 75.27 | 90.41 | 91.13 | 66.78 | 35.62 | 35.87 | 60.15 | 61.24 | 59.05 | 50.39 |
| KD-STDA [11] | 53.17 | 50.95 | 76.76 | 89.24 | 98.66 | 73.76 | 43.75 | 53.45 | 49.04 | 67.23 | 65.56 | 55.81 |
| KA-MCD [16] | 57.96 | 65.26 | 61.66 | 81.75 | 91.79 | 71.68 | **50.85** | 56.73 | 51.01 | 64.18 | 65.95 | 57.74 |
| MLD-DA [12] | 78.16 | 75.49 | 83.34 | 99.86 | 96.83 | 86.74 | 45.36 | 66.17 | 58.37 | 63.87 | 70.71 | 60.90 |
| REDA [9] | 86.70 | 81.08 | 88.98 | 92.35 | 88.77 | 87.58 | 44.07 | 52.01 | 60.14 | 60.46 | 64.67 | 56.27 |
| AAD [15] | 52.50 | 60.00 | 80.86 | 89.84 | 95.99 | 75.84 | 32.71 | 62.92 | 63.34 | 64.46 | 72.15 | 59.12 |
| MobileDA [14] | 76.19 | 58.77 | 83.74 | 97.56 | 97.84 | 82.82 | 41.83 | 57.14 | 59.41 | 64.38 | 61.55 | 56.86 |
| UNI-KD [4] | 78.85 | 82.68 | **92.14** | 97.29 | **99.34** | 90.06 | 44.48 | 60.13 | 62.99 | 71.03 | 72.21 | 62.17 |
| **Ours** | **89.88** | **85.63** | 88.57 | **99.92** | 99.13 | **92.63** | 49.73 | **70.74** | **72.14** | 71.73 | **76.95** | **68.26** |

very marginal and large variance can be observed, indicating the challenge of performing domain adaptation with shallow networks. On the contrary, the compact student trained with our proposed RCD-KD consistently performs better than other methods.

## 4.3 Benchmark with KD-based DA methods

We compare our proposed method with other KD-based DA methods as shown in Table 2 and Table 3. We applied above methods on our teacher-student settings. We also report the performance of pre-trained teacher (generally considered as the upper limit) and the student trained on source domain but tested on target domain (*namely*, Student-Only) as the lower limit. We highlight the best performance with **bold** for each scenario and the averaged performance. Note that this comparison does not include the teacher as it benefits from more complex network architecture. See **Supplementary** for more experimental results of additional transfer scenarios.

Some observations can be found from above two tables. Firstly, compared to Student-Only, all the benchmark methods can obviously improve compact student's generalization on target domain. However, some of them (*e.g.*, KD-STDA in **HAR**, **FD** and **SSC**, KA-MCD in **HAR** and **FD**) even perform worse than directly applying UDA on compact student (*e.g.*, MDDA and SASA in Table 1). The reason is that those methods blindly trust teacher's predictions on target domain as mentioned and learning with such unreliable knowledge will result in inferior performance. Secondly, the methods using source-only teachers (*i.e.*, AAD and MobileDA) failed to achieve better performance than others (*e.g.*, UNI-KD and MLD-DA) which employ teachers trained on both source and target domains. This observation indicates that for cross-domain KD, it is critical for the teacher to possess the knowledge of both domains. Thirdly, introducing the source domain specific knowledge to the student like KD-STDA and MLD-DA apparently cannot guarantee better generalization on target data. Intuitively, the student still needs to pay more attention on target domain or focus on domain-shared knowledge as UNI-KD suggested. Lastly, our proposed method consistently outperforms other benchmarks over all the datasets and achieves the highest score in most of transfer scenarios. Meanwhile, our RCD-KD can even achieve comparable performance as the teacher model in some datasets (*e.g.*, **HAR**, **HHAR** and **FD**) with more compact model architectures. It indicates the effectiveness of transferring the adaptive knowledge via proposed RL-based sample selection module and domain-invariant knowledge via domain discriminator.

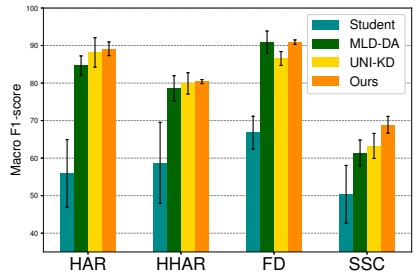

Figure 2: **TCN →1D-CNN**.

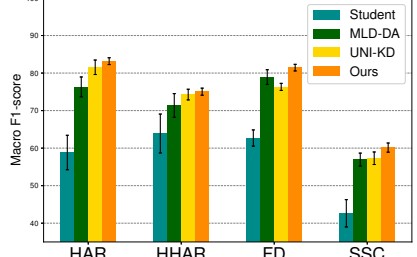

Figure 3: **Resnet-34 → Resnet-18**.

Table 4: Teacher with different UDA methods.

| T-Types | HAR | HHAR | FD | SSC |
|---|---|---|---|---|
| MDDA | 89.16 | 81.65 | 90.45 | 61.62 |
| SASA | 88.16 | 80.39 | 90.83 | 63.25 |
| CoDATS | 93.39 | **82.98** | 91.08 | 66.89 |
| **DANN (ours)** | **94.68** | 82.37 | **92.63** | **68.26** |

Table 5: Framework ablation for proposed method.

| $\mathcal{L}_{KD}$ | $\mathcal{L}_{DC}$ | $\mathcal{L}_{RKD}$ | HAR | HHAR | FD | SSC |
|---|---|---|---|---|---|---|
| ✓ | | | 79.46 | 75.69 | 72.35 | 52.63 |
| | ✓ | | 85.51 | 79.03 | 77.95 | 62.39 |
| ✓ | ✓ | | 89.32 | 78.99 | 89.13 | 60.65 |
| | ✓ | ✓ | **94.68** | **82.37** | **92.63** | **68.26** |

## 4.4 Ablation Study

**Different Teacher-Student (T − S) pairs.** Besides transferring the knowledge from a cumbersome **1D-CNN** teacher to a compact **1D-CNN** student following [4], we also evaluate our proposed method with other **T − S** pairs. Specifically, we adopted a temporal convolutional network (**TCN**) [37] as the teacher and **1D-CNN** as the student, a **Resnet-34** [38] as the teacher and **Resnet-18** as the student as depicted in Fig. 2 and Fig. 3. We can see that our proposed method consistently outperforms other benchmark methods, further indicating its effectiveness under different **T − S** configurations.

**Different Teacher Generation.** As stated in the Methodology section, our proposed method re-utilizes the domain discriminator after teacher's training with DANN. To further demonstrate that our framework can be generalized to other UDA methods, we evaluate our framework with teachers generated by various DA methods as shown in Table 4. Specifically, we utilize two discrepancy-based DA methods, *i.e.*, MDDA [5] and SASA [36] and one additional adversarial-based DA method named CoDATS [8]. Note that for teachers generated from MDDA and SASA, we need to train the domain discriminator as shown in Algorithm 1. On the contrary, for teachers generated from CoDATS and DANN (**ours**), the parameters of discriminator are frozen during student's training. From Table 4, we can see that employing teachers from MDDA and SASA slightly underperforms CoDATS and DANN. The possible reason is that adding additional training steps for domain discriminator will inevitably increase training difficulty in terms of model convergence. But they still perform better than other benchmarks in Tables 2 and 3, indicating that our approach is not limited by the teacher training strategies as long as the teacher possesses the source and target domain knowledge. A well pre-trained domain discriminator is preferred as it can improve the training efficiency.

**Framework Ablation.** There are two key components in our proposed framework: the domain-invariant knowledge transferring via $\mathcal{L}_{DC}$ and the RL-based domain adaptive knowledge transferring via $\mathcal{L}_{RKD}$. We conduct the ablation study to evaluate their contributions. We also include the standard knowledge distillation in which the student is optimized with $\mathcal{L}_{KD}$ as Eq. (1) shows. From Table 5, we can see that coarsely aligning teacher's and student's predictions over all target samples ($\mathcal{L}_{KD} + \mathcal{L}_{DC}$) might lead to negative transferring in some datasets (*e.g.*, **HHAR** and **SSC**), whose performance is inferior to the one only applying $\mathcal{L}_{DC}$. Our proposed method ($\mathcal{L}_{DC} + \mathcal{L}_{RKD}$) can significantly mitigate this issue via the proposed RL-based target sample selection module.

**Reinforced Sample Selection Ablation.** To investigate the contribution of proposed reward and RL-based model selection module, we conduct the ablation study as shown in Table 6. Some observation can be found from above table. Firstly, including all the target samples to train the compact student will inevitable introduce negative transfer, resulting in unsatisfying generalization performance on target domain. Either utilizing the model uncertainty or transferability to explicitly

Table 6: Reinforced sample selection ablation. "Full samples" denotes utilizing whole target samples for KD; '$\mathcal{R}_2$', '$\mathcal{R}_3$' denote directly utilizing proposed uncertainty and transferability for sample selection; '$\mathcal{R}_2^\dagger$', '$\mathcal{R}_3^\dagger$' denote utilizing RL with $\mathcal{R}_2$ and $\mathcal{R}_3$ as reward for sample selection; $(\mathcal{R}_2 + \mathcal{R}_3)^\dagger$ is **our** proposed method.

| Datasets | Full Samples | Partial Samples | | | | | |
|---|---|---|---|---|---|---|---|
| | | $\mathcal{R}_2$ | $\mathcal{R}_2^\dagger$ | $\mathcal{R}_3$ | $\mathcal{R}_3^\dagger$ | $\mathcal{R}_2 + \mathcal{R}_3$ | $(\mathcal{R}_2 + \mathcal{R}_3)^\dagger$ |
| HAR | 89.32 | 91.65 | 93.91 | 92.31 | 93.96 | 93.53 | **94.68** |
| HHAR | 78.99 | 78.30 | 81.73 | 80.33 | 82.29 | 81.04 | **82.37** |
| FD | 89.13 | 90.17 | 91.93 | 89.51 | 91.08 | 91.85 | **92.63** |
| SSC | 60.65 | 63.16 | 62.98 | 65.81 | 67.49 | 65.20 | **68.26** |

Table 7: Comparison of Computational Complexity.

| Methods | KD-STDA | KA-MCD | MLD-DA | REDA | AAD | MobileDA | UNI-KD | **Ours** |
|---|---|---|---|---|---|---|---|---|
| Time (sec) | 1.68 | 4.55 | 1.91 | 1.78 | 0.91 | 1.28 | 3.26 | 16.42 |

select target sample (as shown in column $\mathcal{R}_2$ and $\mathcal{R}_3$) would improve student's performance in most of datasets. Secondly, dynamically selecting target samples using our RL-base module with either of proposed rewards (as shown in column $\mathcal{R}_2^\dagger$ and $\mathcal{R}_3^\dagger$) will further improve student's performance, indicating the effectiveness of RL-based module in mitigating negative transfer. Lastly, combining model uncertainty and transferability as the reward to dynamically select suitable target samples based on student's capacity yields best performance.

**Computational Complexity.**    We performed the time complexity analysis for our method and the results are shown in Table 7. Specifically, we measure the training time for our proposed method and other benchmarks with a NVIDIA 2080Ti GPU. The reported results are measured with one epoch on single transfer scenario on FD dataset, which has the largest training samples (about 1,800 samples per transfer scenario) among evaluated datasets. We can see that our method does require more training time compared to other benchmarks, reflecting its greater complexity. The primary computational costs arise from two factors. The first part is the generation of $K$ historical experiences at each step. This could be significantly reduced by using a smaller K. The second factor is the MCD module which conducts multiple inference processes for uncertainty estimation. This computational burden could be further decreased by adopting alternative uncertainty estimation methods. Nevertheless, although our training time is longer than other benchmarks, we argue that it is still within an acceptable range, especially considering the performance improvement it could bring. Meanwhile, we also evaluate the scalability of our approach to larger dataset. See **Supplementary** for more scalability analysis.

## 5  Conclusion and Limitations

In this paper, we propose a framework for cross-domain knowledge distillation on time series. Specifically, we utilize an adversarial domain discriminator to assist the compact student learn the domain-invariant knowledge from the cumbersome teacher. Meanwhile, we design a reinforcement learning-based target sample selection module to effectively transfer teacher's knowledge which is suitable for compact student. The experimental results demonstrate the effectiveness of our proposed method in enhancing the generalization of compact student on target domain. There are also some limitations for our proposed framework. On the one hand, we still need to pre-train a cumbersome teacher with advanced UDA methods, which involves more training time than others. On the other hand, we only utilize the distance between teacher's and student's logits to assess sample's transferability, which might overlook some intrinsic information from feature space. In the future, we will extend our work to (1) jointly train teacher and student for cross-domain knowledge distillation, and (2) consider feature representations into sample transferability assessment.

## Acknowledgments

This work is supported by the Agency for Science, Technology and Research (A*STAR) Singapore under its NRF AME Young Individual Research Grant (Grant No. A2084c0167) and the National Research Foundation, Singapore under its AI Singapore Programme (AISG2-RP-2021-027).

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

# A  Supplementary

## A.1  Uncertainty Estimation with Monte Carlo Dropout Method

Given an input data set $X = \{x_1, ..., x_n\}$ and the respective outputs $Y = \{y_1, ..., y_n\}$, the conventional machine learning methods intend to find an optimal model $\Phi(x; \boldsymbol{\theta})$, which is parameterized with $\boldsymbol{\theta}$, to map the input $X$ to the $Y$. After training, the optimal model $\Phi(x; \boldsymbol{\theta})$ will give a single point prediction for certain test sample with static $\boldsymbol{\theta}$. On the contrary, the Bayesian methods (*e.g.*, Bayesian neural networks) can generate predictive distributions instead of a single point prediction for estimating model uncertainty. With defining $\Phi(x; \boldsymbol{\theta})$ with a prior $P(\boldsymbol{\theta})$ over parameter space $\boldsymbol{\theta}$, the training objective is then turned to find an optimal posterior distribution over $\boldsymbol{\theta}$:

$$P(\boldsymbol{\theta}|X, Y) = \frac{P(Y|X, \boldsymbol{\theta})P(\boldsymbol{\theta})}{P(Y|X)}. \tag{11}$$

The prediction value of $y$ with input $x$ is the weighted average of model predictions over all possible sets of parameters $\boldsymbol{\theta}$ with various posterior probabilities as Eq. (12) shows.

$$\begin{aligned} P(y|x, X, Y) &= \int P(y|x, \boldsymbol{\theta})P(\boldsymbol{\theta}|X, Y)d\boldsymbol{\theta} \\ &= \mathbb{E}_{\boldsymbol{\theta} \sim P(\boldsymbol{\theta}|X, Y)}[\Phi(x; \boldsymbol{\theta})] \end{aligned} \tag{12}$$

However, the posterior distribution $P(\boldsymbol{\theta}|X, Y)$ is intractable as shown in previous works. Alternatively, Gal and Ghahramani proved that a DNN with arbitrary non-linear depth and dropout is mathematically equivalent to a Bayesian approximation of the probabilistic deep Gaussian process. They proposed a method named Monte Carlo Dropout which utilizes a dropout distribution $\hat{P}(\boldsymbol{\theta})$ to approximate $P(\boldsymbol{\theta}|X, Y)$. To be specific, for the $l$-th layer ($l = 1, ..., L$) in a model with total $L$ layers, the parameter distribution $\boldsymbol{\theta}_l$ is defined as:

$$\boldsymbol{\theta}_l = \mathbf{M}_l * diag([Z_{l,i}]_{i=1}^{D_l}), \tag{13}$$

where $\mathbf{M}_l \in \mathcal{R}^{D_l \times D_{l-1}}$ is a matrix with variational parameters and $diag(\cdot)$ is an operator to map a vector to a diagonal matrix. $Z_{l,i} \sim Bernoulli(q_i)$ is independently sampled from Bernoulli distribution, where $i = 1, ..., D_{l-1}$. $q_i$ is the probability of dropout. Subsequently, the Eq. (12) is reformulated as:

$$\mathbb{E}_{\boldsymbol{\theta} \sim \hat{P}(\boldsymbol{\theta})}[\Phi(x; \boldsymbol{\theta})] \approx \frac{1}{N} \sum_{n=1}^{N} \Phi(x; \boldsymbol{\theta}^n). \tag{14}$$

Practically, Eq. (14) means to enable the dropout of model during test phase and forward $N$ times for each sample $x_i$. Furthermore, for classification tasks we employ the entropy to measure teacher's uncertainty on target data as Eq. (15) shows:

$$\mathcal{H}_i = -\sum_c (\frac{1}{N} \sum_n p(y = c|x_i, \boldsymbol{\theta}^n) log(\frac{1}{N} \sum_n p(y = c|x_i, \boldsymbol{\theta}^n))), \tag{15}$$

where $p(y = c|x_i, \boldsymbol{\theta}^n)$ represents the probability of sample belong to class $c$ and it is the softmax outputs of teacher model $\Phi(x; \boldsymbol{\theta}^n)$ on the $n$-th forward pass. Intuitively, a higher value of the predictive entropy $\mathcal{H}_i$ will be obtained when all classes are predicted to have equal probabilities, which means the teacher is less confident about the specific data.

## A.2  Model Architecture

### A.2.1  Teacher and Student model

As illustrated in Fig. 4(a) and (b), the teacher and student model are constructed with three stacked CNN blocks as the backbone and one fully connected layer as the classifier. Each CNN block consists

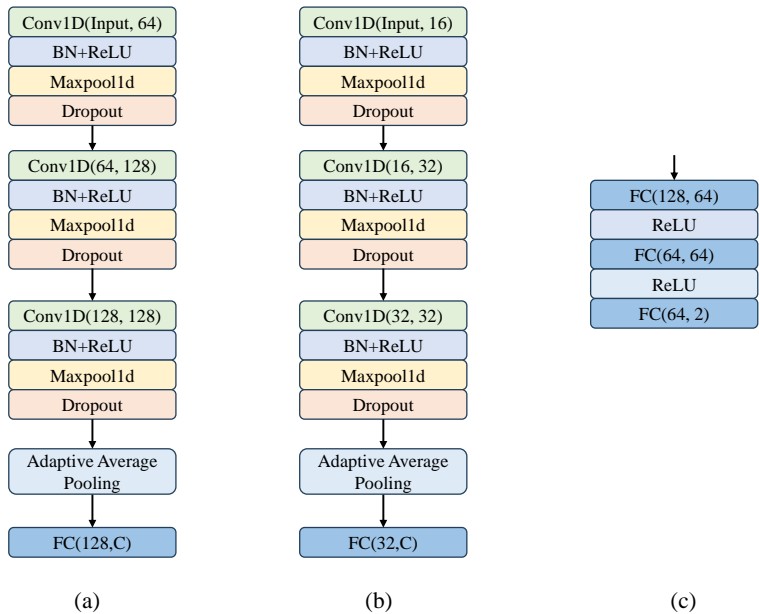

Figure 4: Network Architectures for (a) **1D-CNN** Teacher, (b) **1D-CNN** Student and (c) Domain Discriminator.

of a 1D convolutional layer, followed by a BatchNorm layer, an activation layer (ReLU), a 1D MaxPooling layer and a Dropout layer. Here, 'Conv1D' represents the 1D convolutional layer and the first variable in the bracket represents the number of input channels and the second one represents the number of output channels. 'BN' is a BatchNorm layer. 'FC' represents a fully connected layer. 'C' represents the number of classes.

Meanwhile, to demonstrate the deployment on edge device we compared our 1D-CNN teacher and student from four perspectives as shown in Table 8. Here, we employ Raspberry Pi 3B+ as the edge device for deployment. We can see that the student reduces 15.46 times parameters, 16.98 times FLOPs and 13.54 times memory usage compared to its teacher. Besides, the inference of student is 21.67 times faster than teacher on the edge device. Meanwhile, in our manuscript we have already shown that the student trained with our method is able to achieve comparable performance as the teacher. This enables our compact student to potentially meet the real-time response and on-site deployment requirements for certain time series applications.

Table 8: Comparison of model complexity.

|  | # Para. (M) | # FLOPs (M) | Memory Usage (Mb) | Inference Time (Sec) |
|---|---|---|---|---|
| T | 0.201 | 0.917 | 83.73 | 4.16 |
| S | 0.013 | 0.054 | 6.33 | 0.192 |
| Rate | 15.46× | 16.98× | 13.54× | 21.67× |

### A.2.2 Domain discriminator

Fig. 4(c) is the network architecture of domain discriminator. It consists of three linear layer followed by ReLU activation layers. The output of domain discriminator is a 2-classes probability distribution to indicate whether the feature maps come from the teacher with source domain data as input or from the student with target domain data as input.

### A.2.3 Dueling DDQN

Table 9 presents the details of our dueling DDQN. The left column is the state-value estimation stream and the right column is the advantage estimation column. The 'NoisyFC' represents a linear layer whose weights and biases are perturbed by a parametric function of the noise. The conventional

Table 9: Network Architecture for Dueling DDQN.

| Layers | Dueling DDQN | |
|---|---|---|
| #1 | FC(Input, 1024)+ReLU | |
| #2 | NoisyFC(1024,1024)+ReLU | NoisyFC(1024,1024)+ReLU |
| #3 | NoisyFC(1024,1) +ReLU | NoisyFC(1024, 2)+ReLU |
| #4 | Aggregation | |

linear layer can be expressed as $y = wx + b$, where $w \in \mathbb{R}^{q \times p}$, $b \in \mathbb{R}^q$ are trainable weights and biases, $x \in \mathbb{R}^p$ and $y \in \mathbb{R}^q$ are the inputs and outputs, respectively. In the NoisyNets, the weights and biases are reformulated as Eq. (16) and (17), respectively. Here, $\mu^w \in \mathbb{R}^{q \times p}$, $\sigma^w \in \mathbb{R}^{q \times p}$, $\mu^b \in \mathbb{R}^q$, $\sigma^b \in \mathbb{R}^q$ are the trainable weights and biases. $\odot$ is the element-wise multiplication. $\epsilon^w$ and $\epsilon^b$ are the factorised Gaussian noise, where each entry $\epsilon_{i,j}^w = f(\epsilon_i)f(\epsilon_j)$, $\epsilon_j^b = f(\epsilon_j)$ and $f(x) = sgn(x)\sqrt{|x|}$. Adding such parametric noise to the deep reinforcement learning agent will enhance the efficiency of exploration.

$$w = \mu^w + \sigma^w \odot \epsilon^w \tag{16}$$

$$b = \mu^b + \sigma^b \odot \epsilon^b \tag{17}$$

## A.3    Additional Transfer Scenarios

We evaluate our proposed method on another five additional transfer scenarios on four datasets as shown in Table 10 and Table 11. From above two Tables, we can sse that our proposed method consistently achieves better performance than other SOTA methods, further indicating its effectiveness in transferring the knowledge under the cross-domain scenarios.

Table 10: Marco F1-scores on HAR and HHAR across three independent runs.

| Methods | HAR Transfer Scenarios | | | | | | HHAR Transfer Scenarios | | | | | |
|---|---|---|---|---|---|---|---|---|---|---|---|---|
| | $18 \to 27$ | $20\to5$ | $24\to8$ | $28\to27$ | $30\to20$ | Avg | $0\to2$ | $5\to0$ | $6\to1$ | $7\to4$ | $8\to3$ | Avg |
| Teacher | 98.23 | 90.57 | 97.08 | 100 | 92.21 | 95.62 | 66.56 | 33.25 | 94.47 | 94.99 | 96.68 | 77.19 |
| Student-Only | 98.37 | 48.78 | 77.38 | 61.17 | 76.41 | 72.42 | 61.94 | 27.43 | 69.10 | 77.72 | 80.51 | 63.34 |
| KD-STDA | **100** | 75.77 | 90.77 | 97.77 | 86.36 | 90.13 | 61.93 | 28.04 | 92.65 | 91.33 | 96.30 | 74.05 |
| KA-MCD | 85.22 | 78.03 | 86.14 | 91.19 | 74.28 | 82.97 | 43.90 | 33.35 | 92.32 | 94.27 | 97.02 | 72.17 |
| MLD-DA | 98.82 | 80.57 | 91.90 | **100** | 91.69 | 92.60 | 65.44 | 31.10 | 92.97 | 94.97 | 95.87 | 76.07 |
| REDA | 98.20 | **95.05** | 91.26 | 98.53 | 72.04 | 91.02 | 54.18 | 32.56 | 88.50 | 88.84 | 96.18 | 72.05 |
| AAD | 90.27 | 66.88 | 86.09 | 94.73 | 84.82 | 84.56 | 58.23 | 37.24 | 91.47 | 81.99 | 83.61 | 70.51 |
| MobileDA | 92.86 | 84.96 | 90.45 | 79.12 | 77.56 | 84.99 | 50.27 | 30.83 | 76.12 | 89.70 | 79.25 | 65.23 |
| UNI-KD | **100** | 94.42 | **100** | 92.26 | 87.10 | 94.76 | 62.33 | **39.01** | 92.89 | **96.90** | 96.52 | 77.53 |
| **Proposed** | **100** | 85.26 | 97.92 | **100** | **92.21** | **95.08** | **67.27** | 38.25 | **94.59** | 95.83 | **97.40** | **78.67** |

Table 11: Marco F1-scores on FD and SSC across three independent runs.

| Methods | FD Transfer Scenarios | | | | | | SSC Transfer Scenarios | | | | | |
|---|---|---|---|---|---|---|---|---|---|---|---|---|
| | $1\to0$ | $1\to3$ | $3\to0$ | $3\to1$ | $3\to2$ | Avg | $3\to19$ | $5\to15$ | $6\to2$ | $13\to17$ | $18\to12$ | Avg |
| Teacher | 66.40 | 100 | 62.30 | 100 | 81.65 | 82.07 | 71.85 | 73.69 | 72.21 | 50.74 | 52.96 | 64.29 |
| Student-Only | 45.13 | 91.14 | 44.84 | 93.03 | 70.55 | 68.94 | 43.65 | 41.04 | 48.21 | 38.78 | 47.28 | 43.79 |
| KD-STDA | 47.55 | 93.02 | 51.26 | 99.81 | 76.28 | 73.58 | 61.24 | 66.97 | 67.05 | 43.05 | 49.92 | 57.65 |
| KA-MCD | 51.77 | 98.69 | 48.50 | 93.65 | 83.05 | 75.13 | 61.13 | 63.23 | 70.96 | 39.56 | 46.86 | 56.35 |
| MLD-DA | 51.98 | 99.67 | 52.14 | 96.01 | 75.62 | 75.08 | 66.23 | 70.30 | 69.33 | 44.22 | 44.13 | 57.65 |
| REDA | 53.65 | 96.21 | 52.48 | 96.60 | 80.85 | 75.96 | 56.09 | 61.96 | 53.59 | 40.50 | 36.26 | 49.68 |
| AAD | 46.42 | 94.65 | 52.09 | 98.65 | 87.11 | 75.78 | 62.75 | 64.81 | **71.78** | 44.52 | 49.18 | 58.61 |
| MobileDA | 51.71 | 94.92 | 51.17 | 99.86 | 78.51 | 75.23 | 64.16 | 67.67 | 56.74 | 47.50 | **56.56** | 58.53 |
| UNI-KD | 60.91 | **99.97** | 61.00 | **100** | 87.08 | 81.79 | 66.84 | 70.76 | 65.70 | **50.19** | 49.77 | 60.65 |
| **Proposed** | **61.99** | 99.67 | **62.42** | **100** | **87.76** | **82.37** | **69.49** | **72.73** | 67.01 | 49.31 | 52.52 | **62.21** |

## A.4    Sensitivity Analysis

### A.4.1    Hyper parameter $N$

In our proposed method, one of the key hyper parameters is $N$, which is the number of teachers utilized for calculating the uncertainty on a specific sample. It relates to our reward function, thus

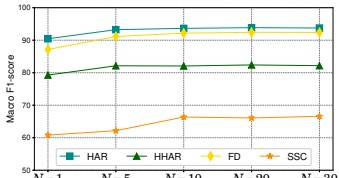

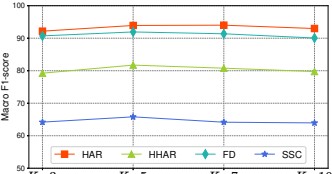

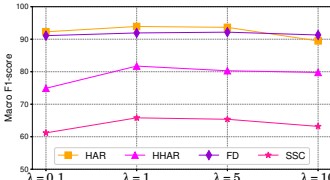

| Figure 5: Sensitivity of $N$. | Figure 6: Sensitivity of $K$. | Figure 7: Sensitivity of $\lambda$. |

Table 12: Sensitivity Analysis on $\alpha_1$ and $\alpha_2$.

| Dataset | $\alpha_1$=0.2 $\alpha_2$=1.8 | $\alpha_1$=0.4 $\alpha_2$=1.6 | $\alpha_1$=0.6 $\alpha_2$=1.4 | $\alpha_1$=0.8 $\alpha_2$=1.2 | $\alpha_1$=1.0 $\alpha_2$=1.0 | $\alpha_1$=1.2 $\alpha_2$=0.8 | $\alpha_1$=1.4 $\alpha_2$=0.6 | $\alpha_1$=1.6 $\alpha_2$=0.4 | $\alpha_1$=1.8 $\alpha_2$=0.2 |
|---------|------|------|------|------|------|------|------|------|------|
| HAR | **94.68** | 94.23 | 94.49 | 93.58 | 93.25 | 92.65 | 92.68 | 92.72 | 92.05 |
| HHAR | **82.37** | 82.35 | 94.49 | 82.10 | 81.59 | 82.11 | 81.04 | 81.25 | 81.34 |
| FD | 92.63 | 92.34 | **92.87** | 92.10 | 91.89 | 91.74 | 92.01 | 91.34 | 92.05 |
| SSC | 68.26 | **68.47** | 68.15 | 67.65 | 67.80 | 66.65 | 66.95 | 66.35 | 66.01 |

affecting the learning process of target sample selection policy. Intuitively, the larger $N$ will lead to more accurate estimation of teacher's uncertainty and provide more accurate reward. As illustrated in Fig. 5, student's performance is gradually increased with $N$ but will keep at some certain level when $N \geq 10$. The larger value of $N$ also increases the total cost in terms of training time. Empirically, $N = 5$ or $N = 10$ are appropriate, and we choose $N = 10$ in our experiments for all the datasets.

### A.5 Hyper parameter $K$

Another hyper parameter in our proposed approach is the episodes length $K$ for generating historical experience and we perform the analysis as illustrated in Fig. 6. From Fig. 6, we can see that our proposed method is not very sensitive to $K$. But a large value of $K$ will result in longer training time. $K = 5$ is sufficient to generate informative historical experience for training the dueling DDQN.

#### A.5.1 Hyper parameter $\lambda$

Regarding hyper parameter $\lambda$ which is to balance the contribution of domain confusion loss $\mathcal{L}_{DC}$ and the distillation loss $\mathcal{L}_{RKD}$, we can see from Fig. 7 that a small value of $\lambda$ (*e.g.* $\lambda = 0.1$) will make the student more focus on learning domain-invariant knowledge from $\mathcal{L}_{DC}$. It will result in worse performance in datasets like **HHAR** and **SSC**. A higher value of $\lambda$ will obviously enhance student's generalization capability on target domain. $\lambda \in [1, 5]$ is a suitable range based on our experiment results.

#### A.5.2 Hyper parameter $\alpha_1$ and $\alpha_2$

To limit our reward within the range of -1 to 1, $\alpha_1$ and $\alpha_2$ should satisfy below constrains: $\alpha_1 \in (0, 2)$, $\alpha_2 \in (0, 2)$ and $\alpha_1 + \alpha_2 = 2$. We perform grid search on four datasets as shown in Table 12. We can see that the student trained with low $\alpha_1$ value and high $\alpha_2$ value can achieve better performance than other configurations, indicating transferability contributes more to the final performance than uncertainty. In all of our experiments, we set $\alpha_1 = 0.2$ and $\alpha_2 = 1.8$ for all datasets for simplicity.

#### A.5.3 Hyper parameter $\tau$

For hyper parameter $\tau$ which is the temperature to soften teacher's logits, we perform the analysis ranged from 1 to 16 as shown in Table 13. We can see that higher value of temperature (e.g., $\tau = 16$) would over-smooth teacher's logits, resulting in poor distillation performance. Generally, $\tau = 2$ or $\tau = 4$ is a good choice for our method.

Table 13: Sensitivity Analysis for temperature $\tau$

| Dataset | $\tau = 1$ | $\tau = 2$ | $\tau = 4$ | $\tau = 8$ | $\tau = 16$ |
|---------|------------|------------|------------|------------|-------------|
| HAR | 92.14 | 94.68 | 94.23 | 91.35 | 89.45 |
| HHAR | 80.14 | 82.37 | 81.45 | 79.41 | 76.49 |
| FD | 90.79 | 92.63 | 92.74 | 88.51 | 85.41 |
| SSC | 65.10 | 67.49 | 66.98 | 63.21 | 59.01 |

Table 14: Comparison with different sample selection strategies in AL.

| Methods | HAR | HHAR | FD | SSC |
|---------|-----|------|-----|-----|
| Baseline | 89.32 | 78.99 | 89.13 | 60.65 |
| LC | 79.21 | 76.22 | 74.14 | 52.9 |
| LC Consist. | 82.01 | 75.43 | 74.45 | 56.11 |
| LC Consist. + RL | 84.9 | 76.24 | 81.45 | 60.01 |
| M | 80.55 | 75.9 | 82.05 | 58.03 |
| M Consist. | 83.55 | 78.91 | 81.9 | 59.45 |
| M Consist. + RL | 90.11 | 80.01 | 80.79 | 61.97 |
| H | 88.31 | 79.09 | 88.18 | 59.23 |
| H Consist. | 91.65 | 78.3 | 90.17 | 63.16 |
| H Consist. + RL | 93.91 | 81.73 | 91.93 | 62.98 |

## A.6 Comparison with sample selection strategies in Active Learning

We conduct the ablation study on three uncertainty-based active learning (AL) strategies, including least confidence (LC), sample margin (M) and sample entropy (H). The results are presented in Table 14. We take student trained with our framework using whole target samples as the baseline (i.e., without RL). 'LC' refers to leveraging student's confidence to directly select samples. 'LC Consist.' refers to using the consistency of teacher's and student's confidence for explicitly sample selection. 'LC Consist. + RL' refers to leveraging 'LC Consist.' as reward to learn optimal sample selection policy. We can see that: firstly, almost all uncertainty-based AL strategies exhibit performance degradation compared to the baseline. This could be attributed to the unreliable uncertainty estimation from student's outputs, especially at early training stage. Additionally, among these strategies, entropy performs the best, likely because it considers the overall probability distribution which might partially address student's unreliable predictions issue. Secondly, utilizing uncertainty consistency instead of uncertainty alone could enhance performance in most settings, as incorporating teacher's knowledge through consistency provides a more reliable measure. Lastly, our RL module could further enhance student's performance via employing any of uncertainty consistency as the reward, indicating its effectiveness.

## A.7 Scalability to Larger Datasets

To verify the efficiency and scalability of our method on larger time series (TS) dataset, we conduct experiments on another Human Activity Recognition dataset named PAMAP2. Table 15 compares the dataset complexity of PAMAP2 with the datasets we employed in our manuscript. Note that it is meaningless to compare the total size of datasets in our settings as our experiment evaluate transfer scenario between single subject. We summarize the averaged number of samples, channels, data length and classes across all transfer scenarios for these datasets. We also report the time complexity of our method across these datasets (i.e., training time per epoch for single transfer scenario). From Table 15, we can see PAMAP2 is larger in terms of number of samples and more complex in terms of number of channels and classes. Compared with FD, the epoch training time for PAMAP2 only increases 2 times as the number of training samples increases about 4 times, indicating that our method scales well in terms of computational efficiency on larger TS dataset.

Meanwhile, we also conduct the performance comparison between our method and benchmarks on PAMAP2 with randomly selected 5 transfer scenarios. The experimental results are summarized as Table 16. We can see that our proposed method consistently outperform other benchmarks in terms of average Macro F1-score, even though it cannot achieve the best performance on some transfer

Table 15: Summary of Datasets.

| Datasets | No. of Samples | No. of Channels | Data Length | No. of Classes | Training Time (Sec) |
|---|---|---|---|---|---|
| HAR | 216 | 9 | 128 | 6 | 1.61 |
| HHAR | 1150 | 3 | 128 | 6 | 9.07 |
| FD | 1828 | 1 | 5120 | 3 | 16.43 |
| SSC | 1428 | 1 | 3000 | 5 | 7.09 |
| PAMAP2 | 8180 | 36 | 256 | 11 | 31.64 |

Table 16: Performance Comparison on PAMAP2.

| Methods | 102→104 | 106→103 | 107→105 | 105→106 | 107→102 | Avg. |
|---|---|---|---|---|---|---|
| KD-STDA | 66.19 | 53.12 | 46.34 | 67.87 | 59.75 | 58.65 |
| KA-MCD | 34.35 | 49.16 | 49.92 | 33.95 | 35.97 | 40.67 |
| MLD-DA | 68.14 | 50.85 | 61.23 | 75.03 | 63.32 | 63.71 |
| REDA | **71.49** | 53.31 | 59.11 | 74.75 | **64.86** | 64.70 |
| AAD | 60.28 | 51.61 | 48.01 | 73.64 | 45.55 | 55.82 |
| MobileDA | 67.14 | 54.09 | **64.21** | 74.67 | 63.35 | 64.69 |
| UNI-KD | 64.82 | **70.82** | 43.92 | 69.65 | 56.20 | 59.28 |
| **Proposed** | 68.33 | 68.86 | 59.56 | **75.44** | 62.50 | **66.96** |

scenarios. This observation indicates that the effectiveness of our proposed method can also be generalized to larger time series dataset.

