# OpenReview forum: "Reinforced Cross-Domain Knowledge Distillation on Time Series Data"
_NeurIPS.cc/2024/Conference — NeurIPS 2024 poster_

### Official Review · Reviewer_nLZW · 2024-07-11

**Soundness:** 3
**Presentation:** 2
**Contribution:** 2
**Rating:** 4
**Confidence:** 3

**Summary:**

The authors motivate the work by identifying limitations in existing approaches that integrate knowledge distillation into domain adaptation frameworks. Specifically, they note that coarsely aligning outputs over all source and target samples neglects the network capacity gap between teacher and student models, leading to poor distillation efficiency.
The proposed RCD-KD framework uses an adversarial discriminator module to align teacher and student representations between source and target domains. It also formulates target sample selection as a reinforcement learning problem, using a novel reward function based on uncertainty consistency and sample transferability. A dueling Double Deep Q-Network (DDQN) is employed to learn the optimal selection policy.

**Strengths:**

Well-motivated: The authors clearly articulate the limitations of existing approaches and provide a compelling rationale for their proposed method.
Comprehensive evaluation: The experimental results are extensive, covering four different datasets across various time series tasks. The comparisons with state-of-the-art methods are thorough and demonstrate consistent improvements.

**Weaknesses:**

Theoretical foundation: While the paper provides a detailed description of the proposed method, it lacks a strong theoretical foundation or analysis. Adding theoretical insights or guarantees would strengthen the contribution.
Computational complexity: The paper does not provide a detailed analysis of the computational complexity of the proposed method compared to existing approaches. Given the focus on resource-constrained devices, this information would be valuable.
Hyperparameter sensitivity: Although some hyperparameter settings are provided, a more comprehensive analysis of the method's sensitivity to different hyperparameters would be beneficial.
Limited discussion on failure cases: While the paper shows impressive results, a more in-depth discussion of scenarios where the method might fail or underperform would provide a more balanced view.

**Questions:**

How does the computational complexity of RCD-KD compare to existing methods, especially considering the reinforcement learning component?
Have you explored the performance of the method on longer time series or datasets with a larger number of classes? How does it scale?
The paper mentions using DANN to pre-train the teacher model. How sensitive is the performance to the choice of the teacher's pre-training method?
How does the method perform when there is a significant domain shift between source and target domains? Are there cases where the performance degrades significantly?
Have you considered extending the approach to handle multi-source domain adaptation scenarios?

**Limitations:**

The authors acknowledge some limitations of their work in the conclusion section. They note that pre-training a cumbersome teacher with advanced UDA methods involves more training time than other approaches. Additionally, they mention that using only the distance between teacher's and student's logits to assess sample transferability might overlook intrinsic information from the feature space.
These are valid limitations, and it's commendable that the authors have included them. However, the discussion could be expanded to include potential implications of these limitations and possible strategies to address them in future work.

---

> ### Author Rebuttal · Authors · 2024-08-06
>
> Thanks to Reviewer nLZW for all the comments.
>
> **Response to Weak-1 (Theoretical Foundation):** We appreciate the suggestion to include a stronger theoretical foundation or analysis to further strengthen the contribution. While we acknowledge the value that theoretical insights and guarantees could bring to the paper, our primary focus in this work has been on the practical implementation and empirical validation of our method. Our goal was twofold: to tackle the prevalent issues of domain shift and model complexity in time series applications, and to demonstrate the superior effectiveness of our approach compared to existing benchmarks through extensive experiments on a range of real-world time series tasks. We believe that the empirical success of our method is also very crucial for research community, and thus, in this paper we emphasizes empirical results over theoretical analysis.
>
> Nonetheless, we will definitely consider incorporating a theoretical foundation in future extensions of this research. Potential directions include: (1) Theoretical Justification for RL: include a theoretical analysis of using reinforcement learning to optimize sample selection, examining the exploration-exploitation trade-off and its impact on performance. (2) Uncertainty Consistency Analysis: explore the theoretical relationship between uncertainty consistency and model performance, supported by mathematical proofs and derivations.
>
> **Response to Weak-2 and Q-1 (Computational Complexity):** Please kindly refer to **Response to Major Weak-2** for Reviewer **Ltvb** for algorithm complexity analysis and **Response to Weak-1-2** for Reviewer **bzJ9** for model complexity analysis.
>
> **Response to Weak-3 (Hyperparameter Sensitivity):** Please kindly refer to  **Response to Weak-3** for Reviewer **E4vw** for sensitivity analysis of hyper parameters.
>
> **Response to Weak-4 (Failure Case Discussion):** Indeed, there are some cases where our approach does not perform as expected, such as $2\to7$ in **HHAR** and $0\to11$ in **SSC**. The poor adaptation performance in these scenarios may be attributed to the larger domain shift, as even the complex teacher model and other benchmarks struggle to achieve good adaptation. When there is a significant domain gap that the complex teacher model cannot effectively address, the target knowledge it provides may also be unreliable for the student. To explore this potential failure cause, we conducted further validation on another HAR dataset characterized by larger domain gaps (See **Response to Q-4**) and we are willing to incorporate a detailed discussion in the revised manuscript.
>
> **Response to Q-2 (Scalability to Larger Dataset):** Please kindly refer to **Response to Weak-2** for Reviewer **E4vw** for the analysis of our method on larger dataset.
>
> **Response to Q-3 (Teacher Sensitivity):**  As shown in Table 4 of our manuscript, we evaluate our framework with teachers pre-trained with different DA methods. Specifically, we utilize two discrepancy-based DA method (i.e., MDDA and SASA) and one adversarial-based DA method named CoDATS. The difference is that: for teachers pre-trained with discrepancy-based DA method, we have to train the domain discriminator. But for teachers pre-trained with adversarial-based DA methods, we can obtain the well-performed discriminator during teacher's training process. Thus, during student's training, the parameters of discriminator are frozen. From Table 4 of our manuscript, we can see that employing teachers from MDDA and SASA slightly underperforms CoDATS and DANN. The possible reason is that adding additional training steps for domain discriminator will inevitably increase training difficulty in terms of model convergence. But they still perform better than other benchmark methods. It indicates that our approach is not limited by the teacher pre-training strategies.
>
> **Response to Q-4 (Larger Domain shift):** To evaluate our method under larger domain shift, we conducted some preliminary experiments on another dataset called HPP HAR [15]. This dataset includes 12 subjects, each performing 6 activities, with the smartphone placed in three different positions: pants, a shirt, and a backpack. The results are shown in the table below. As observed, almost all methods experience performance degradation when faced with a significant domain gap. A potential reason for this is that these KD-based cross-domain methods (including ours) are highly dependent on the teacher's performance. If the teacher fails to capture domain-invariant representations, the student model may also be negatively impacted. Due to time constraints, we were unable to explore other SOTA domain adaptation methods to enhance the teacher's performance and verify the above hypothesis. We plan to conduct this verification in future extensions of our research.
>
> Scenario|Teacher|Student-Only|MobileDA|UNI_KD|Ours
> :---:|:---:|:---:|:---:|:---:|:---:|
> S01Pants $\to$ S02Backpack|43.54|23.45|37.41|35.45|38.95
> S01Shirt $\to$ S03Pants|56.10|33.45|49.41|41.12|47.77
>
> **Response to Q-5 (Extent to Multi-Source):** Some adjustments may be necessary if we extend our approach to multi-source domain adaptation (MSDA) scenarios. Firstly, instead of utilizing RL to select proper target samples, in MSDA we could leverage RL to dynamically select suitable source domains (or source samples), which are most relevant to target domain, to minimize the negative transfer as much as possible. Secondly, some of the key components in RL need to be re-defined. For instance, we need to re-formulate reward function to offer essential feedback to DDQN.  A potential solution could be some functions to measure the data distribution discrepancy between source and target domain. Lastly,  for MSDA, the teacher should be the expert who globally predicts well on the mixture of source domains. Methods like STEM [16] could be possible solutions for training a proper teacher in multi-source domain scenario.

---

### Official Review · Reviewer_bzJ9 · 2024-07-11

**Soundness:** 2
**Presentation:** 3
**Contribution:** 2
**Rating:** 5
**Confidence:** 3

**Summary:**

This paper proposes a knowledge distillation method for unsupervised domain adaptation models in time series classification. After pre-training the teacher model with existing domain adaptation methods, the proposed Reinforced Cross-Domain Knowledge Distillation (RCD-KD) method selects suitable target domain samples for knowledge distillation with reinforcement learning and distills knowledge from the pre-trained teacher model to a smaller student model. Empirical experimental results on four public time series datasets demonstrate the effectiveness of the proposed method over other state-of-the-art benchmarks.

**Strengths:**

* The writing and presentation of this paper are good. The setup of the proposed problem and the proposed method are described clearly.

* This paper proposes a new distillation method. It makes some contributions in using reinforcement learning for target sample selection in knowledge distillation.

* Experiments in four public datasets show the effectiveness of the proposed method, which outperforms some domain adaptation and knowledge distillation methods. There are also some ablation studies to validate the designs of the reward and the training losses.

**Weaknesses:**

* The proposed problem seems to be a simple two-stage combination of domain adaptation and knowledge distillation and may not be realistic enough, which does not show the significance of solving them together in one problem. Besides, authors claim that it can help ‘on edge devices with very limited computational resources’, but the experiments only distill from one bigger CNN to a smaller CNN, which does not make a difference in enabling deployment on edge devices.

* Authors should explain more on why using reinforcement learning to select samples works better than directly using designed metrics such as uncertainty and transferability. Reinforcement learning will add a lot of computation costs to the training process and it is unclear what the learned selection policy looks like. It seems that the proposed reward cannot solve the claimed issue that ‘in the cross-domain scenario, teacher’s knowledge on each individual target sample may not be always reliable’. Besides, this paper proposes a distillation method for time series data, but the method does not show its special designs for time series.

* The domain discrimination loss is confusing. How does it enable ‘transfer the domain-invariant knowledge’ to the student model? If the teacher model already learns domain-invariant knowledge, why don’t we achieve this by distilling teacher features to the student model?

**Questions:**

In addition to the Weaknesses, there are some minor points of suggestion:

* Figure 1 is a little hard to understand, especially the relations between the reward module and other parts.

* There are some typos, such as Line 118 ‘we consider (the) target sample selection task as a Markov Decision Process which can (be) addressed by reinforcement learning.’

**Limitations:**

Authors discussed the limitations in the conclusion part.

---

> ### Author Rebuttal · Authors · 2024-08-06
>
> Thanks to Reviewer bzJ9 for all the comments.
>
> **Response to Weak-1-1 (Simple Combination):** We clarify that our solution is not a simple two-stage combination of DA and KD. Particularly, we optimize DA loss $L_{DC}$ and KD loss $L_{RKD}$ together, addressing domain shift and model complexity simultaneously. Moreover, we consider the following two solutions as simple two-stage combination. 1. KD$\to$DA:  first distill the knowledge from teacher to student on source data, and then directly perform DA on student. 2. DA$\to$KD: first perform DA on the teacher and then conduct distillation. As shown in below table, these two solutions perform better than Source-Only model, but significantly worse than ours. The reason is that if performing KD on source data first, followed by DA, the student would become biased towards the source data during KD stage. And due to its compact network architecture, the student's adaptation performance would also be affected. If we perform DA first and then perform KD on target data, it would result in poor distillation efficiency in KD stage due to the lack of ground truth on target data. These issues  are consistent with findings from previous research [12][13].
>
> Methods|HAR|HHAR|FD|SSC
> :---:|:---:|:---:|:---:|:---:
> Student|55.94|58.74|66.78|50.39
> KD$\to$DA|73.66|72.67|67.87|54.26
> DA$\to$KD|74.29|74.54|70.40|60.08
> Ours|94.68|82.37|92.63|67.49
>
> **Response to Weakness-1-2 (Model Complexity):** To demonstrate the deployment on edge device, we compared our 1D-CNN teacher and student from four perspectives as shown in below table. Here, we employ Raspberry Pi 3B+ as the edge device for deployment. We can see that the student reduces 15.46 times parameters, 16.98 times FLOPs and 13.54 times memory usage compared to its teacher. Besides, the inference of student is 21.67 times faster than teacher on the edge device. Meanwhile, in our manuscript we have already shown that the student trained with our method is able to achieve comparable performance as the teacher. This enables our compact student to potentially meet the real-time response and on-site deployment requirements for certain time series applications.
>
> ||# Para.(M)|# FLOPs(M)|Memory Usage (Mb)|Inference Time (Sec)
> :---:|:---:|:---:|:---:|:---:|
> T|0.201|0.917|83.73|4.16
> S|0.013|0.054|6.33|0.192
> Rate|15.46$\times$|16.98$\times$|13.54$\times$|21.67$\times$
>
> **Response to Weak-2-1 (Why RL works):** We argue that the primary reason why RL performs better is due to its inherent balance between exploitation and exploration. If we solely exploit the gained knowledge (i.e., directly using uncertainty or transferability for sample selection), the student is likely to become stuck at certain sub-optimal points (see comparison of $R_2$, $R_2^\dagger$,  $R_3$ and $R_3^\dagger$ in Table 6 of our manuscript). The exploration in RL allows the agent to explore new possibility from the environment. In our implementation, we utilize the ‘NoisyFC’ whose weights and biases are perturbed by a parametric function of the noise to enhance the efficiency of exploration. Meanwhile, we also agree that RL does involve more computational costs than others, but the total training time is still acceptable, especially considering the performance improvement it could bring (see **Response to Major Weak-2** for Reviewer **bzJ9**).
>
> **Response to Weak-2-2 (Claimed Issue and No Special Design for TS):** We'd like to clarify that our method does not intend to improve teacher's reliability but to adaptively transfer its target knowledge with our RL-based framework. The unreliability of teacher's target knowledge is the underlying reason why existing approaches that simply integrate KD with UDA frameworks often experience unsatisfactory adaptation performance. Thus, we are highly motivated to utilize RL to select samples aligning with student’s capability to mitigate such unreliable knowledge. Besides, we also agree that there is no special architecture design for TS in our proposed framework. We choose to evaluate on TS as model complexity and domain shift issues are very common in TS. As it is general, we will explore to extend our method to other research areas like CV and NLP in the future.
>
> **Response to Weak-3 (Domain-invariant Knowledge):** The domain discrimination loss originates from Reference [14], which is closely related to adversarial learning. The main idea is to pit two networks against each other. The student is expected to generate invariant representations for both domains and the discriminator is expected to fail to distinguish them. By minimizing this loss, the student would generate similar representations on target domain as the teacher on source domain. In other words, the domain-invariant knowledge would be transferred from teacher to student.
>
> Besides, there are two reasons why we cannot directly transfer the domain-invariant knowledge from teacher to student. First, the compact student model has very limited capacity and cannot capture the same fine-grained patterns in the data as teacher. Coarsely aligning their feature maps, as done in KD-STDA and MLD-DA, would lead to sub-optimal performance on target domain. Secondly, instead of focusing on learning domain-invariant representations, our objective is to improve student's generalization on target domain via teacher's knowledge, which motivates us to adaptively transfer teacher’s target knowledge based on student’s capability. Our ablation study results on framework (see Table 5 of manuscript) also suggest that the performance improvement from domain-invariant loss is very marginal. Conversely, our proposed RKD loss can significantly improve student's generalization capability on target domain.
>
> **Response to Q-1 (Reward not clear):** We will try to improve the clarity of our framework via providing more description in updated version (See Updates in global **Author Rebuttal**).
>
> **Response to Q-2 (Typos):** We will improve the typos in updated version.

---

> > ### Comment · Reviewer_bzJ9 · 2024-08-12
> > **Response to the rebuttal**
> >
> > Thanks for the response. Some of my concerns have been addressed. Additionally, I suggest that some more discussions on RL as well as its training complexity and stability may help, considering that RL is integrated with distillation and domain adaptation to form a complicated training method. It makes sense to me that RL helps explore some new possibilities. But RL may also cause some issues such as unstable training and may not always succeed in exploring and exploiting. Would this lead to some failure cases? What are the differences between the learned policy and designed metrics, does the learned policy show some special properties?
> > Besides, since the problem and the technical design are not limited to time series, it may be better to consider going beyond this specific type of data in the writing and experiments.

---

> > > ### Author Response · Authors · 2024-08-13
> > > **Response to Reviewer's feedback**
> > >
> > > Thanks to Reviewer **bzJ9** for the response to our rebuttal.
> > >
> > > **Response to Q-1 (Include more discussion on RL):**
> > > Thank you for above suggestion. As we responded in our previous rebuttal, we will include the discussion of training complexity in our updated version. Meanwhile, we will also provide more discussion on training stability as shown below.
> > >
> > > >Particularly, a dueling DDQN is employed to learn the optimal target sample selection policy. The dueling architecture can effectively mitigate the risk of overestimation by separately estimating the state value and advantage function, which improves the accuracy of action-value predictions. Meanwhile, to tackle the instability issue often encountered in training deep reinforcement learning models, we leverage strategies such as target network and experience replay. Specifically, the target network provides more stable targets for updating the Q-values by maintaining a separate, slowly updated network for generating target values, while experience replay enables the model to learn from a diverse set of past experiences, further enhancing stability and convergence during training.
> > >
> > > **Response to Q-2 (Unstable training issue of RL):**
> > > In our proposed framework, we utilize the dueling DDQN architecture [1], which was initially developed to address potential instability issues by separating the value and advantage functions. The value stream establishes a solid baseline from which the advantages of actions are evaluated. It helps reduce the risk of overestimating active values and enhances the selection process. Additionally, during training, we incorporate techniques like target network updates and experience replay to further tackle stability concerns (see **Response to Q-1** for more details). Previous research [2] [3] also demonstrated that these techniques can effectively improve the stability of training in deep Q-networks.
> > >
> > >     [1] Wang, Z., Schaul, T., etc. Dueling network architectures for deep reinforcement learning. In ICML  2016.
> > >     [2] Wu, K., Wu, M., Yang., etc. Deep Reinforcement Learning Boosted Partial Domain Adaptation. In IJCAI 2021.
> > >     [3] Pan, J., Wang, etc.. Multisource transfer double DQN based on actor learning. IEEE TNNLS, 2018.
> > >
> > > **Response to Q-3 (Failure Cases Discussion):**
> > > Thank you very much for above comment. Indeed, we have observed some cases where our approach does not perform as expected. One potential factor for poor adaptation performance may be attributed to the larger domain shift (Please kindly refer to our **Response to Weak-4** and **Q-4** for Reviewer **nLZW**). A significant distribution gap between the source and target domains can render the knowledge from the teacher unreliable. However, it is worth noting that most benchmark methods also suffer from performance degradation in the presence of such domain gaps, while our approach consistently shows superior results. This issue might be partially addressed by enhancing the teacher's performance with SOTA domain adaptation techniques. Additionally, another potential cause for failure in exploration and exploitation could be related to the initialization of the dueling DDQN and its optimization trajectory. To address this, we conducted all experiments three times with different random seeds and reported the average performance.
> > >
> > > **Response to Q-4 (Comparison of learned policies and their properties):** Thank you very much for raising this question. Unlike fields such as computer vision, the interpretation of time series data is not straightforward, making it challenging to visualize and directly compare the differences between learned policies and their properties. Currently, we assess these policies solely based on their impact on the student's performance. However, it might become feasible to conduct more explicit comparisons using  synthetic time series data. By manually simulating domain shifts in such data, we could observe how the learned policies select samples and compare their behaviors more directly.
> > >
> > > **Response to Q-5 (Go beyond time series data):** Thank you very much for your valuable feedback. We fully agree with your suggestion to consider extending our method beyond time series data. Our focus on time series in this paper is driven by the following factor. Our group’s research expertise is centered on time series analysis. Hence, we have only tested its effectiveness across multiple time series datasets/tasks. It would be premature for us to claim its general applicability to other fields such as computer vision or natural language processing without further validation.
> > >
> > > Currently, we are very willing to conduct experiments on data types beyond time series. However, due to time constraints, we may not be able to generate sufficient or convincing results. As we mentioned in our rebuttal, we are very interested in exploring these extensions in a journal version if possible and assessing the effectiveness of our approach in different domains.

---

> ### Comment · Reviewer_bzJ9 · 2024-08-13
> **Response to authors**
>
> Thanks for the response. I suggest detailed discussions on RL and more analysis of learned policies and failure cases (such as related experimental analysis and visualization) be included in the paper. I have updated the rating.

---

> > ### Author Response · Authors · 2024-08-13
> >
> > Dear Reviewer,
> >
> > Thank you very much for the time and efforts dedicated to reviewing our paper.  We are so grateful for your constructive suggestion and updating the rating for our paper.

---

### Official Review · Reviewer_E4vw · 2024-07-12

**Soundness:** 3
**Presentation:** 3
**Contribution:** 3
**Rating:** 6
**Confidence:** 3

**Summary:**

This paper proposes a Reinforced Cross-Domain Knowledge Distillation (RCD-KD) framework for time series data, aiming to effectively transfer knowledge from a cumbersome teacher model to a compact student model across different domains. The RCD-KD framework leverages an adversarial domain discriminator to learn domain-invariant features and a reinforcement learning-based sample selection module to dynamically select informative target samples for knowledge distillation. The proposed method significantly improves the student model's generalization ability on the target domain compared to conventional knowledge distillation and domain adaptation techniques.

**Strengths:**

1. By incorporating an adversarial domain discriminator, the framework can effectively learn domain-invariant features, enabling the student model to generalize better to the target domain.
2. The reinforcement learning module dynamically selects informative target samples based on the student model's capacity and uncertainty, mitigating negative transfer and improving the efficiency of knowledge distillation.
3. Extensive experiments on four public datasets across three tasks demonstrate that the proposed RCD-KD framework consistently outperforms other state-of-the-art domain adaptation and knowledge distillation methods.

**Weaknesses:**

1. The framework utilizes the distance between teacher and student logits to assess sample transferability, potentially overlooking intrinsic information from the feature space. More comprehensive discussions on different feature distances could further enhance sample selection.
2. While the experiments demonstrate the effectiveness of the proposed RCD-KD framework on several datasets, it is unclear how well the method would scale to larger and more complex time series datasets with higher dimensional feature spaces. The computational efficiency and scalability of the framework under such settings remain to be investigated.
3. The paper acknowledges that grid search was used to tune hyperparameters such as α1, α2, λ, and τ. However, the sensitivity of the framework's performance to these hyperparameters is not thoroughly analyzed. The framework's robustness to different hyperparameter configurations could potentially limit its practical applicability without extensive tuning.

**Questions:**

See the weaknesses.

**Limitations:**

See the weaknesses.

---

> ### Author Rebuttal · Authors · 2024-08-06
>
> Thanks to Reviewer E4vw for all the comments.
>
> **Response to Weak-1 (Feature Knowledge for Transferability Assessment):** Although we have discussed above weakness as one of our limitations in manuscript, we conducted some preliminary experiments as suggested. We investigate several feature-based KD methods to assess sample transferability on **SSC** as below table shows. Specifically, we utilized L2 [7], attention maps (AT) [8], CORAL [9], and probabilistic knowledge transfer (PKT) [10] to measure the distance between teacher's and student's feature maps. These feature distances are then used to calculate reward  $\mathcal{R}_3$ in Eq.(3). From the table, we can see that transferability assessments using feature distance obviously underperform logits-based method in our proposed framework. The potential reason is that: in our current framework, we utilize MCD module to generate $N$ teachers and then average their logits for calculating uncertainty and transferability. However, unlike logits, each data point in the feature maps carries different meaning depending on its spatial location within its feature space. Simply averaging feature maps across multiple teachers appears unreasonable and impractical, potentially resulting in poor performance. Therefore, the feature-based knowledge cannot be directly integrated into our existing framework. A more comprehensive designs need to be carefully considered.
>
> Methods|0$\to$1|12$\to$5|16$\to$1|7$\to$18|9$\to$14|Avg
> |:---:|:---:|:---:|:---:|:---:|:---:|:---:
> L2|22.73|24.75|47.00|64.23|62.59|44.26
> AT|29.93|38.25|59.56|59.02|56.52|48.66
> CORAL|34.06|38.87|50.22|54.32|52.94|46.08
> PKT|28.20|57.70|47.74|55.70|74.84|52.84
> Logits (**Ours**)|48.39|69.56|70.52|72.20|76.79| 67.49
>
> **Response to Weak-2 (Scalability to Larger Dataset):** To verify the efficiency and scalability of our method on larger time series (TS) dataset, we conduct experiments on another Human Activity Recognition dataset named PAMAP2 [11]. Below table compares the dataset complexity of PAMAP2 with the  datasets we employed in our manuscript. Note that it is meaningless to compare the total size of datasets in our settings as our experiment evaluate transfer scenario between single subject. We summarize the averaged number of samples, channels, data length and classes across all transfer scenarios for these datasets. We also report the time complexity of our method across these datasets (i.e., training time  per epoch for single transfer scenario). From this table, we can see PAMAP2 is larger in terms of number of samples and more complex in terms of number of channels and classes. Compared with **FD**, the epoch training time for PAMAP2 only increases 2 times as the number of training samples increases about 4 times, indicating that our method scales well in terms of computational efficiency on larger TS dataset.
>
> Datasets|No.of Samples|No.of Channels|Data Length|No.of Classes|Training Time (sec)
> :---:|:---:|:---:|:---:|:---:|:---:
> HAR|216|9|128|6|1.61
> HHAR|1150|3|128|6|9.07
> FD|1828|1|5120|3|16.43
> SSC|1428|1|3000|5|7.09
> PAMAP2|8180|36|256|11|31.64
>
> Meanwhile, we also conduct the performance comparison between our method and benchmarks on PAMAP2 with randomly selected 5 transfer scenarios. The experimental results are summarized as below table. We can see that our proposed method consistently outperform other benchmarks in terms of average Macro F1-score, even though it cannot achieve the best performance on some transfer scenarios. This observation indicates that the effectiveness of our proposed method can also be generalized to larger time series dataset.
>
> Methods|102$\to$104|106$\to$103|107$\to$105|105$\to$106|107$\to$102|Avg
> :---:|:---:|:---:|:---:|:---:|:---:|:---:|
> KD-STDA|66.19|53.12|46.34|67.87|59.75|58.65
> KA-MCD|34.35|49.16|49.92|33.95|35.97|40.67
> MLD-DA|68.14|50.85|61.23|75.03|63.32|63.71
> REDA|**71.49**|53.31|59.11|74.75|**64.86**|64.70
> AAD|60.28|51.61|48.01|73.64|45.55|55.82
> MobileDA|67.14|54.09|**64.21**|74.67|63.35|64.69
> UNI-KD|64.82|**70.82**|43.92|69.65|56.20|59.28
> **RCD-KD**|68.33|68.86|59.65|**75.44**|62.50|**66.96**
>
> **Response to Weak-3 (Hyperparameter Sensitivity):** Due to paper space constraints, our sensitivity analysis for hyperparameters $\alpha_1$, $\alpha_2$, $\lambda$, $N$ and $K$ were included in the Supplementary (see Fig. 2,3,4 and Table 5). Please refer to our submitted Supplementary for the details.
>
> For hyperparameter $\tau$ which is the temperature to soften teacher's logits, we added additional analysis ranged from 1 to 16 as shown in below Table. We can see that higher value of temperature (e.g., $\tau = 16$) would over smooth teacher's logits, resulting in poor distillation performance. Generally, $\tau = 2$ or  $\tau = 4$ is a good choice for our method.
>
> Dataset|$\tau=1$|$\tau=2$|$\tau=4$|$\tau=8$|$\tau=16$
> :---:|:---:|:---:|:---:|:---:|:---:
> HAR|92.14|94.68|94.23|91.35|89.45
> HHAR|80.14|82.37|81.45|79.41|76.49
> FD|90.79|92.63|92.74|88.51|85.41
> SSC|65.10|67.49|66.98|63.21|59.01

---

> > ### Comment · Reviewer_E4vw · 2024-08-14
> >
> > Thank you! The authors have addressed all my concerns.

---

> > > ### Author Response · Authors · 2024-08-14
> > >
> > > Dear Reviewer **E4vw**,
> > >
> > > We are very glad to hear that all of your concerns have been addressed.  We really appreciate the time and efforts you have dedicated for providing valuable feedback on our paper.

---

### Official Review · Reviewer_Ltvb · 2024-07-12

**Soundness:** 4
**Presentation:** 4
**Contribution:** 4
**Rating:** 8
**Confidence:** 4

**Summary:**

This paper introduces a reinforcement learning-based active learning method designed to dynamically select target data for knowledge-transfer, whose goal is to bridge the network capacity gap between teacher and student networks within a domain adaptation framework incorporating knowledge distillation. Specifically, the authors propose a novel reward mechanism aimed at learning an optimal policy for selecting target data, considering the student network's capacity.

Another novel aspect of this paper is its successful demonstration of the framework's effectiveness on time-series data. This modality is less explored in both domain adaptation and knowledge distillation research fields.

##################################### Post Rebuttal #####################################

All of my concerns have been properly addressed, and my questions have been answered during rebuttal. I am happy to raise my score to ***Strong Accept***.

##################################### Post Rebuttal #####################################

**Strengths:**

1. The paper is well-written and easy to follow.

2. The proposed reward function for selecting target data in domain adaptation from a large network to a smaller one is novel. This method integrates principles from active learning, reinforcement learning, knowledge distillation, and domain adaptation.

3. The authors offer a detailed description of their implementation in the paper, along with the accompanying code. This makes it straightforward for practitioners to apply their methods or build upon their work.

4. Formulating network output entropy as a reward function for data selection represents a novel approach that effectively incorporates concepts from reinforcement learning into a classification model.

5. The performance lift of the proposed method is very significant in time-series data, which promotes the interests of active-learning-based domain adaptation.

**Weaknesses:**

### Majors:

1. Using model output entropy as a measure of uncertainty level is a common technique in active learning. While applying this to promote uncertainty consistency between teacher and student networks is novel, it is important to acknowledge the context of active learning within the paper. It would be not good to borrow ideas without attribution.

2. Based on my experience, running such a reinforced loop for data selection is particularly time-consuming, especially concerning the Markov chain state update mentioned from Line 127 to 134. Therefore, it would be beneficial for the authors to conduct a time complexity analysis of their proposed method.

### Minors:

1. I believe active learning is highly related to the proposed work. Conducting a literature review on active learning would offer practitioners valuable context, enhancing their understanding of the proposed research.

2. In Line 112, "divergence" would be a more appropriate term for describing the difference between distributions, rather than using "distance".

**Questions:**

1. For Lines 90 to 91, the authors stated that the estimated entropy from the domain-shared feature extractor is deemed unreliable for the student. Could the authors provide further elaboration on this assertion? Specifically, what does this unreliability entail and what are its underlying causes?

2. While DDQN isn't the method proposed in this paper, it should have significant impact on the overall performance. I am curious whether other active learning methods might outperform MCD. I recommend conducting an ablation study to examine how various active learning strategies influence overall performance.

---

> ### Author Rebuttal · Authors · 2024-08-06
>
> Thanks to Reviewer Ltvb for all the comments.
>
> **Response to Major Weak-1 (Comparison with Active Learning):** We fully agree that our method is closely related to AL, particularly in selecting critical samples with uncertainty. We are also willing to include the acknowledge of AL in our updated version. Meanwhile, we'd like to clarify that there are also some distinguished differences between our approach and uncertainty-based AL sample selection methods. Firstly, as noted by the reviewer, our method introduces a novel concept by promoting uncertainty to uncertainty consistency. We use sample uncertainty consistency to assess the student's learning capability, rather than directly filtering samples based on their uncertainty. Additionally, this only forms part of our reward function, as we also incorporate sample transferability to evaluate student's network. Secondly, unlike AL, which explicitly leverages uncertainty for sample selection, our approach employs a RL-based module to heuristically learn the optimal target sample selection policy. And our experimental results on reinforced sample selection ablation (Table 6) have demonstrated its superior capability of performance enhancement compared to directly selecting samples as done in AL. Lastly, the selected target samples with our method will not be labeled as AL. Instead, they will continue to be evaluated without labelling after student's state is updated.
>
> **Response to Major Weak-2 (Computational Complexity):** We performed the time complexity analysis as suggested and the results are shown in below table. Specifically, we measure the training time for our proposed method and other benchmarks with a NVIDIA 2080Ti GPU. The reported results are measured with one epoch on single transfer scenario on **FD** dataset, which has the largest training samples (about 1,800 samples per transfer scenario) among evaluated datasets.
>
> Methods|Time(sec)|Macro F1-score
> :---:|:---:|:---:
> KD-STDA|1.68|55.81
> KA-MCD|4.55|57.74
> MLD-DA|1.91|60.90
> REDA|1.78|56.27
> AAD|0.91|59.12
> MobileDA|1.28|56.86
> UNI-KD|3.26|62.17
> **Ours**|16.42|67.49
>
> We can see that our method does require more training time compared to other benchmarks, reflecting its greater complexity. The primary computational costs arise from two factors. The first part is the generation of $K$ historical experiences at each step, which accounts for approximately 73\% of total training time. This could be significantly reduced by using a smaller K. The second factor, which consumes about 21\% of total training time, is the MCD module which conducts multiple inference processes for uncertainty estimation. This computational burden could be further decreased by adopting alternative uncertainty estimation methods. Nevertheless, although our training time is longer than other benchmarks, we argue that it is still within an acceptable range, especially considering the performance improvement it could bring.
>
> **Response to Minor Weak-1 (Literature Review):** As suggested, we performed a literature review on uncertainty-based active learning and will include it in our updated version (See Updates in global **Author Rebuttal**).
>
> **Response to Minor Weak-2 (Typo):**  We would like to revise it in the final revision.
>
> **Response to Q-1 (Unreliability):** As stated in UNI-KD, the authors employ a data-domain discriminator to estimate sample uncertainty, with inputs derived from feature extractor of compact student trained on both source and target domain (i.e., domain-shared). However, our experiments, which directly applied SOTA UDA methods to compact student (see Table 1 of our manuscript), demonstrated that the compact student cannot fully capture the fine-grained patterns in source and target domains as effectively as teacher. Due to its limited capacity, the student's adaptation performance is inferior, making the uncertainty estimated by UNI-KD unreliable. In contrast, our proposed method leverages more robust teacher to estimate uncertainty via MCD module. Our experimental results further demonstrate its effectiveness on enhancing student’s performance on target domain.
>
> **Response to Q-2 (Ablation with AL):** We conduct the ablation study on three uncertainty-based active learning (AL) strategies, including least confidence (LC), sample margin (M) and sample entropy (H). The results are presented in below table. We take student trained with our framework using whole target samples as the baseline (i.e., without RL).  'LC'  refers to leveraging student's confidence to directly select samples. 'LC Consist.' refers to using the consistency of teacher's and student's confidence for explicitly sample selection.  'LC Consist. +RL' refers to leveraging 'LC Consist.' as reward to learn optimal sample selection policy.
>
> Methods|HAR|HHAR|FD|SSC
> :---|:---:|:---:|:---:|:---:
> Baseline|89.32|78.99|89.13|60.65
> LC|79.21|76.22|74.14|52.9
> LC Consist.|82.01|75.43|74.45|56.11
> LC Consist.+RL|84.9|76.24|81.45|60.01
> M|80.55|75.9|82.05|58.03
> M Consist.|83.55|78.91|81.9|59.45
> M Consist.+RL|90.11|80.01|80.79|61.97
> H|88.31|79.09|88.18|59.23|
> H Consist.|91.65|78.3|90.17|63.16|
> H Consist.+RL|93.91|81.73|91.93|62.98|
>
> We can see that: firstly, almost all uncertainty-based AL strategies exhibit performance degradation compared to the baseline. This could be attributed to the unreliable uncertainty estimation from student's outputs, especially at early training stage. Additionally, among these strategies, entropy performs the best, likely because it considers the overall probability distribution which might partially address student's unreliable predictions issue. Secondly, utilizing uncertainty consistency instead of uncertainty alone could enhance performance in most settings, as incorporating teacher's knowledge through consistency provides a more reliable measure. Lastly, our RL module could further enhance student's performance via employing any of uncertainty consistency as the reward, indicating its effectiveness.

---

> ### Comment · Reviewer_Ltvb · 2024-08-12
> **Comments after Reading Authors' Response**
>
> Thank you to the authors for the detailed response. All of my concerns have been properly addressed, and my questions have been answered. I am happy to raise my score to ***Strong Accept***.

---

> > ### Author Response · Authors · 2024-08-13
> >
> > Dear Reviewer **Ltvb**,
> >
> > We are very glad to hear that your concerns have been addressed. Thank you very much for the time and efforts you have dedicated to reviewing our paper.

---

### Author Rebuttal · Authors · 2024-08-06

## Summary

We sincerely thank all the reviewers for their insightful and valuable feedback. We are pleased that the reviewers recognized the novelty of our work and appreciated our motivation for designing a reinforcement learning-based sample selection approach for cross-domain knowledge transfer. Additionally, we are glad that the comprehensive experimental evaluation across four public datasets was noted, highlighting the superior performance and robustness of our approach. We have carefully considered all the suggestions and comments. The overall summary of our rebuttal is as follows.

* We performed the computational complexity analysis as suggested by Reviewer **Ltvb**,**bzJ9** and **nLZW**.
* We verified the scalability of our method to larger dataset as suggested by Reviewer **E4vw** and **nLZW**,  and scalability to dataset with significant domain shift as suggested by Reviewer **nLZW**.
* We conducted the ablation study with additional active learning strategies as suggested by Reviewer **Ltvb**,
* We added sensitivity analysis for temperature hyper parameter as suggested by **E4vw**, **nLZW**.
* We included experiments on integrating feature distances for sample selection as suggested by Reviewer **E4vw**.
* We provided additional details and clarifications as requested by all reviewers.

## Updates to manuscript:

* Include complexity analysis and additional hyperparameter sensitivity analysis as suggested by Reviewer **Ltvb**,**bzJ9** and **nLZW**.
* Include experimental results on larger dataset PAMAP2 as suggested by Reviewer **E4vw** and **nLZW**.
* Include literature review for active learning as suggested by Reviewer **Ltvb** as follows:
    >Meanwhile, our work also relates to active learning (AL) field specifically in terms of selecting the most critical instances from unlabeled data. Note that here we only discuss the uncertainty-based sampling strategies in active learning as other query strategies (e.g., instance correlation) are beyond the scope of our paper. Particularly, the uncertainty can be measured by three metrics: least confidence, sample margin, and sample entropy [1]. The least confidence methods like [2][3] select the instances which have the highest posterior probability and the margin sampling leverage the margin between posterior probabilities of the first two most probable classes [4]. Unlike above two, the entropy metric measures the uncertainty over the whole output prediction distribution [5][6]. In our method, instead of explicitly utilizing entropy-based uncertainty as AL methods, we propose to leverage the consistency between teacher' and student's entropy-based uncertainty to learn the optimal sample selection policy with dueling DDQN. The experimental results across various time series datasets demonstrate the effectiveness of our method.

* Update caption for Fig.1 and include more description to better illustrate our reward module in our framework as commented by Reviewer **bzJ9**.
>As illustrated in Fig.1, the reward function consists of three parts. The first one is the action $a_k$ which is the output of dueling DDQN. The second part is the uncertainty consistency, estimated by entropy from student's logits $\boldsymbol{q}^S$ and the averaged logits $\overline{\boldsymbol{p}}^T$ of $N$ teachers generated from MCD module. The third part is the sample transferability based on the KL divergence between $\boldsymbol{q}^S$ and $\overline{\boldsymbol{p}}^T$. The output of reward module $r_k$ then will be utilized for the optimization of dueling DDQN for learning optimal sample selection policy.

## References in Rebuttal:

    [1] Tharwat, A., Schenck, W. (2023). A survey on active learning: State-of-the-art, practical challenges and research directions. Mathematics.
    [2] Culotta A, McCallum A. Reducing labeling effort for structured prediction tasks. In: Proceedings of the 20th AAAI 2005
    [3] Zhu J, Wang H, Tsou B, Ma M (2010) Active learning with sampling by uncertainty and density for instances annotations. IEEE Trans Audio Speech Lang Process 18(6)
    [4] Campbell C, Cristianini N, Smola A. Query learning with large margin classifiers. In: Proceedings of the 17th ICML 2000
    [5] Burl MC, Wang E. Active learning for directed exploration of complex systems. In: Proceedings of the 26th (ICML 2009)
    [6] Kim J, Song Y, et al . MMr-based active machine learning for bionamed entity recognition. In: Human language technology and the North American association for computational linguistics
    [7]  A. Romero, N. Ballas, et al., “Fitnets: Hints for thin deep nets,” in ICLR, 2015.
    [8]. Zagoruyko and N. Komodakis, “Paying more attention to attention: Improving the performance of convolutional neural networks via attentiontransfer,” 5th ICLR, 2017.
    [9] B. Sun and K. Saenko, “Deep coral: Correlation alignment for deep do-main adaptation,” in European conference on computer vision. Springer,2016
    [10] N. Passalis and A. Tefas, “Learning deep representations with probabilistic knowledge transfer,” in Proceedings of the ECCV, 2018,
    [11] Reiss, A., Stricker, D. Introducing a new benchmarked dataset for activity monitoring. In 2012 16th international symposium on wearable computers.
    [12] Atif Belal, Madhu Kiran, et al. Knowledge distillation methods for efficient unsupervised adaptation across multiple domains. Image and Vision Computing.
    [13] Granger, E., Kiran, M.. Joint progressive knowledge distillation and unsupervised domain adaptation. In 2020 IJCNN.
    [14] Tzeng, E., Hoffman, J., Saenko, K. Adversarial discriminative domain adaptation. In Proceedings of ICCV (2017).
    [15] Chen, Zhenghua, et al .:Smartphone sensor-based human activity recognition using feature fusion and maximum full a posteriori. IEEE TIM (2019)
    [16] Nguyen, V. A., Nguyen, T., et al. (2021). Stem: An approach to multi-source domain adaptation with guarantees. In Proceedings of ICCV.

---

### Decision · Program_Chairs · 2024-09-25

**Decision:**

Accept (poster)

**Comment:**

Summary:
The paper presents a novel RL-based active learning approach that dynamically selects target data used for transferring knowledge from the teacher network to the student network. The proposed framework is sensitive to the differences in the capacities of the teach and student networks, and its goal is to bridge this gap within a domain adaptation framework incorporating knowledge distillation. To accomplish this, the authors proposed a reward mechanism that learns an optimal policy for selecting data from the target domain, considering the student network’s capacity. The proposed model significantly improves the student model’s generalizability on the target domain compared to standard knowledge distillation and domain adaptation techniques.

Strengths:
* A novel method to select appropriate target data during domain adaptation from a large network to a smaller one. The potential impact of the proposed approach is significant.
* A well-written, well-organized, and an easy to follow paper.
* Experiments are fairly rigorous and are backed up by a detailed description of their implementation, accompanied by the code.
* The performance gains in comparison with the baselines are also significant on the time-series datasets considered.

Weaknesses:
* Because the proposed method incorporates the RL loop during knowledge transfer, it will result in increased training time. While this is not a significant burden on the data sets used in the paper (as shown in their rebuttal), it could become a substantial burden in more complex settings.
* The writing of the paper could be improved by focusing more on attribution to previous techniques/work that is closely related to the proposed techniques.

Overall Recommendation:
This paper presents a novel and elegant RL-based approach for knowledge transfer from the teacher network to the student network under the domain shift scenario. The method's experimental validation is thorough and convincing. The authors did a great job addressing the reviewers' concerns in their rebuttals. Since disseminating the ideas presented in this work can potentially benefit the broader machine-learning community, the recommendation is to accept this paper for NeurIPS.